# Robust Mixture Learning when Outliers Overwhelm Small Groups

**Daniil Dmitriev**[1][*]    **Rares-Darius Buhai**[1][*]    **Stefan Tiegel**[1]    **Alexander Wolters**[2]

**Gleb Novikov**[3]    **Amartya Sanyal**[4]    **David Steurer**[1]    **Fanny Yang**[1]

[1]ETH Zurich    [2]TU Munich
[3]Lucerne School of Computer Science and Information Technology    [4]University of Copenhagen
[*]*Equal contribution*

## Abstract

We study the problem of estimating the means of well-separated mixtures when an adversary may add arbitrary outliers. While strong guarantees are available when the outlier fraction is significantly smaller than the minimum mixing weight, much less is known when outliers may crowd out low-weight clusters – a setting we refer to as list-decodable mixture learning (LD-ML). In this case, adversarial outliers can simulate additional spurious mixture components. Hence, if all means of the mixture must be recovered up to a small error in the output list, the list size needs to be larger than the number of (true) components. We propose an algorithm that obtains order-optimal error guarantees for each mixture mean with a minimal list-size overhead, significantly improving upon list-decodable mean estimation, the only existing method that is applicable for LD-ML. Although improvements are observed even when the mixture is non-separated, our algorithm achieves particularly strong guarantees when the mixture is separated: it can leverage the mixture structure to partially cluster the samples before carefully iterating a base learner for list-decodable mean estimation at different scales.

## 1   Introduction

Estimating the mean of a distribution from empirical data is one of the most fundamental problems in statistics. The mean often serves as the primary summary statistic of the dataset or is the ultimate quantity of interest that is often not precisely measurable. In practical applications, data frequently originates from a mixture of multiple groups (also called subpopulations) and a natural goal is to estimate the distinct means of each group separately. For example, we might like to use representative individuals to study how a complex decision or procedure would impact different subpopulations. In other applications, such as genetics [1] or astronomy [2] research, finding the means themselves can be a crucial first step towards scientific discovery. In both scenarios, the algorithm should output a list of estimates that are close to the unobservable true means.

However, in practice, the data may also contain outliers, for example due to measurement errors or abnormal events. We would like to find good mean estimates for all inlier groups even when the proportion of such *additive adversarial contaminations* is larger than some smaller groups that we want to properly represent. The central open question that motivates our work is thus:

*What is the cost of efficiently recovering small groups that may be outnumbered by outliers?*

More specifically, consider a scenario where the practitioner would like to recover the means of small but significant enough inlier groups which constitute at least $w_{\mathrm{low}} \in (0, 1)$ proportion of the

38th Conference on Neural Information Processing Systems (NeurIPS 2024).

(corrupted) data. If $k$ is the number of such inlier groups, for all $i \in [k]$, we then denote by $w_i \geqslant w_{\text{low}}$ the unknown weight of the $i-$th group with mean $\mu_i$. Further, we use $\varepsilon$ to refer to the proportion of additive contamination – the data that comes from an unknown adversarial distribution. The goal is to estimate the unknown means $\mu_i$ for all $i \in [k]$.

Existing works on robust mixture learning such as [3, 4] consider the problem when the fraction of additive adversarial outliers is smaller than the weight of the smallest subgroup, i.e. $\varepsilon < w_{\text{low}}$. However, for large outlier proportions where $\varepsilon \geqslant w_{\text{low}}$, these algorithms are not guaranteed to recover small clusters with $w_i \leqslant \varepsilon$. In this case, outliers can form additional spurious clusters that are indistinguishable from small inlier groups. As a consequence, generating a list of size equal to the number of components would possibly lead to neglecting the means of small groups. In order to ensure that the output contains a precise estimate for each of the small group means, it is thus necessary the estimation algorithm to provide a list whose size is strictly larger than the number of components. We call this paradigm *list-decodable mixture learning* (LD-ML), following the footsteps of a long line of work on list-decodable learning (see Sections 2 and 5).

Specifically, the main challenge in LD-ML is to provide a *short* list that contains good mean estimates for all inlier groups. We first note that there is a minimum list size the algorithm necessarily has to output to guarantee that all groups are recovered. For example, consider an outlier distribution that includes several copies of the smallest inlier group distribution with means spread out throughout the domain. Since inlier groups are indisntinguishable from spurious outlier ones, the shortest list that includes means of all inlier groups must be of size at least $|L| \geqslant k + \frac{\varepsilon}{\min_i w_i}$. Here, $\frac{\varepsilon}{\min_i w_i}$ can be interpreted as the minimal list-size overhead that is necessary due to "caring" about groups with weight smaller than $\varepsilon$. The key question is hence how good the error guarantees of an LD-ML algorithm can be when the list size overhead stays close to $\frac{\varepsilon}{\min w_i}$, while being agnostic to $w_i$ aside from the knowledge of $w_{\text{low}}$. Furthermore, we are interested in *computationally efficient* algorithms for LD-ML, especially when dealing with high-dimensional data.

To the best of our knowledge, the only existing efficient algorithms that are guaranteed to recover inlier groups with weights $w_i \leqslant \varepsilon$ are *list-decodable mean estimation* (LD-ME) algorithms. LD-ME algorithms model the data as a mixture of one inlier and outlier distribution with weights $\alpha \leqslant 1/2$ and $1 - \alpha$ respectively. Provided with the weight parameter $\alpha$, they output a list that contains an estimate close to the inlier mean with high probability. However, for the LD-ML setting, the inlier weights $w_i$ are not known and we would have to use LD-ME algorithms with $w_{\text{low}}$ as weight estimates for each group. This leads to suboptimal error in particular for large groups, that hence (somewhat counter intuitively) would have to "pay" for the explicit constraint to recover small groups. Furthermore, even if LD-ME were provided with $w_i$, by design it would treat inlier points from other components also as outliers, unnecessarily inflating the fraction of outliers to $1 - w_i$ instead of $\varepsilon$.

**Contributions**    In this paper, we propose an algorithm that (i) correctly estimates the weight of each component only given a lower bound and (ii) does not overestimate proportion of outliers when components are well-separated. In particular, we construct a meta-algorithm that uses mean estimation algorithms as base learners that are designed to deal with adversarial corruptions. This meta-algorithm inherits guarantees from the base learner and any improvement of the latter translates to better results for LD-ML. For example, if the base learner runs in polynomial time, so does our meta-algorithm. Our approach of using the output of weak base learners to achieve better performance is reminiscent of the *boosting* paradigm that is common in machine learning practice.

Our algorithm achieves significant improvements in error and list-size guarantees for multiple settings. For ease of comparison, we summarize error improvements for inlier Gaussian mixtures in Table 1 (see Theorem 3.3 for the general result regarding distributions with bounded moments). The main focus of our contributions is represented in the second row; that is the setting where outliers outnumber some inlier groups with weight $w_j \leqslant \varepsilon$ and the inlier components are *well-separated*, i.e., $\|\mu_i - \mu_j\| \gtrsim^1 \sqrt{\log \frac{1}{w_{\text{low}}}}$, where $\mu_i$'s are the inlier component means. As we mentioned before, robust mixture learning algorithms, such as [4, 7], are not applicable here and the best error guarantees in prior work is achieved by an LD-ME algorithm, e.g. from [3]. While its error bounds are of order $O(\sqrt{\log \frac{1}{w_{\text{low}}}})$ for a list size of $O(\frac{1}{w_{\text{low}}})$, our approach guarantees error $O(\sqrt{\log \frac{\varepsilon}{w_i}})$ for a list size

---

[1]We adopt the following standard notation: $f \lesssim g$, $f = O(g)$, and $g = \Omega(f)$ mean that $f \leqslant Cg$ for some universal constant $C > 0$. $\widetilde{O}$-notation hides polylogarithmic terms.

| Type of inlier mixture | Best prior work | Ours | Inf.-theor. lower bound |
|---|---|---|---|
| Large ($\forall j : \varepsilon \leqslant w_j$), sep. groups | $\widetilde{O}(\varepsilon/w_i)$ | $\widetilde{O}(\varepsilon/w_i)$ | $\Omega(\varepsilon/w_i)$, see [5] |
| Small ($\exists j : \varepsilon \geqslant w_j$), sep. groups | $O\left(\sqrt{\log \frac{1}{w_{\text{low}}}}\right)$ | $O\left(\sqrt{\log \frac{\varepsilon+w_i}{w_i}}\right)$ | $\Omega\left(\sqrt{\log \frac{\varepsilon+w_i}{w_i}}\right)$, Prop. 3.5 |
| Non-separated groups | $O\left(\sqrt{\log \frac{1}{w_{\text{low}}}}\right)$ | $O\left(\sqrt{\log \frac{1}{w_i}}\right)$ | $\Omega\left(\sqrt{\log \frac{1}{w_i}}\right)$, see [6] |

Table 1: For a mixture of Gaussian components $\mathcal{N}(\mu_i, I_d)$, we show upper and lower bounds for the **error of the $i$-component** given a output list $L$ (of the respective algorithm) $\min_{\hat{\mu} \in L} \|\hat{\mu} - \mu_i\|$. When the error doesn't depend on $i$, all means have the same error guarantee irrespective of their weight. Note that depending on the type of inlier mixture, different methods in [3] are used as the 'best prior work': robust mixture learning for the first row and list-decodable mean estimation for the rest.

of $k + O(\frac{\varepsilon}{w_{\text{low}}})$. Remarkably, we obtain the same error guarantees as if an oracle would run LD-ME on each inlier group *with the correct weight* $w_i$ separately (with outliers). Hence, the only cost for recovering small groups is the increased list-size overhead of order $O(\frac{\varepsilon}{w_{\text{low}}})$. Further, a sub-routine in our meta-algorithm also obtains novel guarantees under *no* separation assumption, as shown in the third row of Table 1. This algorithm achieves the same error guarantees for similar list size as a base learner that knows the correct weights of the inlier components.

Based on a reduction argument from LD-ME to LD-ML, we also provide information-theoretic (IT) lower bounds for LD-ML. If the LD-ME base learners achieve the IT lower bound (possible for inlier Gaussian mixtures), so does our LD-ML algorithm. In synthetic experiments, we implement our meta-algorithm with the LD-ME base learner from [8] and show clear improvements compared to the only prior method with guarantees, while being comparable or better than popular clustering methods such as k-means and DBSCAN for various attack models.

## 2 Settings

We now introduce the learning settings that appear in the paper. Let $d \in \mathbb{N}_+$ be the ambient dimension of the data and $k \in \mathbb{N}_+$ be the number of mixture components (inlier groups/clusters).

### 2.1 List-decodable mixture learning under adversarial corruptions

We focus on mixtures that consist of distributions that are sufficiently bounded in the following sense.
**Definition 2.1.** Let $t \in \mathbb{N}_+$ be even and let $D(\mu)$ be a distribution on $\mathbb{R}^d$ with mean $\mu$. We say that $D(\mu)$ has *sub-Gaussian $t$-th central moments* if for all even $s \leqslant t$ and for every $v \in \mathbb{R}^d$ with $\|v\| = 1$, $\mathbb{E}_{x \sim D} \langle x - \mu, v \rangle^s \leqslant (s-1)!!$.

This class of distributions is closely related to commonly studied distributions in the literature (see, e.g., [5]) with bounded $t$-th moment. Our requirement for the boundedness of all moments $s \leqslant t$ stems from the fact that our algorithm should adapt to unknown and possibly non-uniform mixture weights.

We assume that we are given samples from a corrupted $d$-dimensional mixture of $k$ inlier distributions $D_i(\mu_i)$ satisfying Definition 2.1, where the mixture is defined as

$$\mathcal{X} = \sum_{i=1}^{k} w_i D_i(\mu_i) + \varepsilon Q, \tag{2.1}$$

and $\sum_{i=1}^{k} w_i + \varepsilon = 1$, where for all $i = 1, \ldots, k$, it holds that $w_i \geqslant w_{\text{low}}$. Further, an $\varepsilon > 0$ proportion of the data comes from an *outlier* distribution $Q$ chosen by the adversary with full knowledge of our algorithm and inlier mixture. Samples drawn from $D_i(\mu_i)$ constitute the $i^{\text{th}}$ *inlier cluster*. The goal in mixture learning under corruptions as in Eq. (2.1), is to design an algorithm that takes in i.i.d. samples from $\mathcal{X}$ and outputs a list $L$, such that for each $i \in [k]$, there exists $\hat{\mu} \in L$ with small estimation error $\|\mu_i - \hat{\mu}\|$.

To the best of our knowledge, we are the first to study the *list-decodable mixture learning* problem (LD-ML) that considers the case of large fractions of outliers $\varepsilon \geqslant \min_i w_i$ and the goal is to achieve

small estimation errors while the list size $|L|$ remains small. While in robust estimation problems, the fractions of inliers and outliers are usually provided to the algorithm, in mixture learning, the mixture proportions are explicit quantities of interest. Throughout the paper, we hence assume that *both* the true weights $w_i$ of the mixture and the fraction of outliers $\varepsilon$ are *unknown*. Instead, by definition in Eq. (2.1), we assume knowledge of a valid lower bound $w_{\text{low}} \leqslant \min_i w_i$.

Note that when $\varepsilon \lesssim \min_i w_i$, the problem is known as robust mixture learning and can be solved with list size $|L| = k$ as discussed in [3, 4, 7]. However, algorithms for robust mixture learning fail when the fraction of outliers becomes comparable to the inlier group size. In the presence of "spurious" adversarial clusters, it is information-theoretically impossible to output a list $L$, such that (i) $|L| = k$ and (ii) $L$ contains precise estimate for each true mean.

## 2.2   Mean estimation under adversarial corruptions

In order to solve LD-ML, we use mean estimation procedures that have provable guarantees under adversarial contamination. Mean estimation can be viewed as a particular case of the mixture learning problem in Eq. (2.1) with $k = 1$, the fraction of inliers $\alpha = w_1$ and the fraction of outliers $\varepsilon = 1 - \alpha$. The mean estimation algorithms we use to solve LD-ML with $w_{\text{low}}$ need to exhibit guarantees under a stronger adversarial model, where the adversary can also replace a small fraction (depending on $w_{\text{low}}$) of the inlier points; see details in Definition B.1. This is a special case of the general contamination model as opposed to the slightly more benign additive contamination model in Eq. (2.1). For different regimes of $\alpha$ we use black-box learners that solve corresponding regime when *provided with $\alpha$*.

**Robust mean estimation**   When the majority of points are inliers, we are in the RME setting. Robust statistics has studied this setting with different corruption models and efficient algorithms are known to achieve information-theoretically optimal error guarantees (see Section 5).

**List-decodable mean estimation**   When inliers form a minority, we are in the list-decodable setting and are required to return a list instead of a single estimate. We refer to this setting as cor-kLD (*corrupted known list-decoding*). For mixture learning, $\alpha$ is usually unknown and we need to solve the cor-aLD (*corrupted agnostic list-decoding*) problem (i.e., $\alpha$ is *not provided*, but instead a lower bound $\alpha_{\text{low}} \in [w_{\text{low}}, \alpha]$ is given to the algorithm). Finally, when only additive adversarial contamination is present, as in Eq. (2.1), we recover the standard list-decoding setting studied in prior works (see Section 5) that we call sLD (*simple list-decoding*). In Appendix G we show that two algorithms designed for sLD also exhibit guarantees for cor-kLD for any $w_{\text{low}}$.

# 3   Main results

We now present our main results for list-decodable and robust mixture learning defined in Section 2. In Section 3.1, we provide algorithmic upper bounds and information-theoretic lower bounds. For the special case of spherical Gaussian mixtures, we show in Section 3.1 that we achieve optimality. Our results are constructive as we provide a meta-algorithm for which these bounds hold.

As depicted in Figure 1, our meta-algorithm (Algorithm 2) is a two-stage process. The outer stage (Algorithm 6) reduces the problem to mean estimation by leveraging the mixture structure and splitting the data into a small collection $\mathcal{T}$ of sets $T$. Each set $T \in \mathcal{T}$ should (i) contain at most one inlier cluster (and few samples from other clusters) and (ii) the total number of outliers across all sets should be at most $O(\varepsilon n)$. We then run the inner stage (Algorithm 3) on sets $T$, which outputs a mean estimate for the inlier cluster in $T$. First, a cor-aLD algorithm identifies the weight of the inlier cluster and returns the result of a cor-kLD base learner with this weight. Then, if the weight is large, we improve the error via an RME base learner. A careful filtering procedure in both stages achieves the significantly reduced list size and better error guarantees. We require the base learners to satisfy the following set of assumptions.

**Assumption 3.1** (Mean-estimation base learners for mixture learning)**.**   Let $t$ be an even integer and consider the corruption setting defined in Definition B.1. Further, let the inlier distribution $D(\mu^*) \in \mathcal{D}$ where $\mathcal{D}$ is the the family of distributions satisfying Definition 2.1 for $t$. We assume that

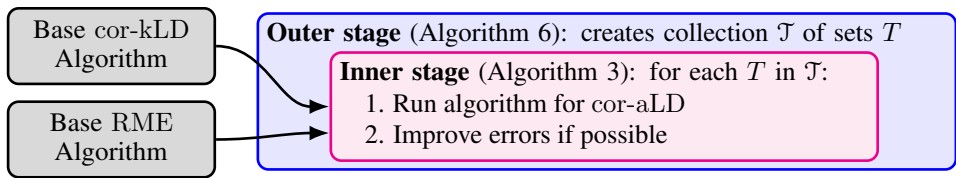

Figure 1: Schematic of the meta-algorithm (Algorithm 2) underlying Theorem 3.3

(a) for $\alpha \in [w_{\text{low}}, 1/3]$ in the cor-kLD regime, there exists an algorithm $\mathcal{A}_{\text{kLD}}$ that uses $N_{LD}(\alpha)$ samples and $T_{LD}(\alpha)$ time to output a list of size bounded by $1/\alpha^{O(1)}$ that with probability at least $1/2$ contains some $\hat{\mu}$ with $\|\hat{\mu} - \mu^*\| \leqslant f(\alpha)$, where $f$ is non-increasing.

(b) for $\alpha \in [1 - \varepsilon_{\text{RME}}, 1]$, with $0 \leqslant \varepsilon_{\text{RME}} \leqslant 1/2 - 2w_{\text{low}}^2$ in the RME regime, there exists an RME algorithm $\mathcal{A}_R$ that uses $N_R(\alpha)$ samples and $T_R(\alpha)$ time to output with probability at least $1/2$ some $\hat{\mu}$ with $\|\hat{\mu} - \mu^*\| \leqslant g(\alpha)$, where $g$ is non-increasing.

Note that the sample and time-complexity functions such as $N_{LD}$ and $T_{LD}$, might depend on $t$, for example growing as $d^t$. We emphasize that (i) the guarantees of our meta-algorithm depend on the guarantees of the base learners and (ii) we only require the base learners to work in the well-studied setting with *known* fraction of inliers. Corollary 3.4 uses known base learners for Gaussian distributions achieving information-theoretically optimal error bounds. There also exists base learners for distributions beyond Gaussians, such as bounded covariance or log-concave distributions, see, e.g. [9].

### 3.1 Upper bounds for list-decodable mixture learning

Key quantities that appear in our error bounds are the relative proportion of inliers $\tilde{w}_i$ and outliers $\tilde{\varepsilon}_i$:

$$\tilde{w}_i = \frac{w_i}{w_i + \varepsilon + w_{\text{low}}^2} \quad \text{and} \quad \tilde{\varepsilon}_i = 1 - \tilde{w}_i. \tag{3.1}$$

These quantities reflect that each set $T$ in the inner stage contains at most one inlier cluster and a small ($\lesssim w_{\text{low}}^2$) fraction of points from other inlier clusters. We now present a simplified version of our main result in Theorem 3.3 (see Theorem C.1 for the detailed result) that allows for a more streamlined presentation of the results using the following 'well-behavedness' of $f$ and $g$.

**Assumption 3.2.** Let $f, g$ be as defined in Assumption 3.1. For some $C > 0$, we assume (i) $\varepsilon_{\text{RME}} \geqslant 0.01$, (ii) $\forall x \in (0, 1/3], f(x/2) \leqslant Cf(x)$, and (iii) $\forall x \in [0.99, 1], g(x - (1-x)^2) \leqslant Cg(x)$.

We are now ready to state the main result of the paper.

**Theorem 3.3.** *Let $d, k \in \mathbb{N}_+$, $w_{\text{low}} \in (0, 1/2]$, and $t$ be an even integer. Let $\mathcal{X}$ be a $d$-dimensional mixture distribution following Eq. (2.1). Let $\mathcal{A}_{\text{kLD}}$ and $\mathcal{A}_R$ satisfy Assumptions 3.1 and 3.2 for some even $t$. Further, suppose that $\|\mu_i - \mu_j\| \gtrsim \sqrt{t}(1/w_{\text{low}})^{4/t} + f(w_{\text{low}})$ for all $i \neq j \in [k]$.*

*Then there exists an algorithm that, given $\text{poly}(d, 1/w_{\text{low}}) \cdot (N_{LD}(w_{\text{low}}) + N_R(w_{\text{low}}))$ i.i.d. samples from $\mathcal{X}$ as well as $d, k, w_{\text{low}}$, and $t$, runs in time $\text{poly}(d, 1/w_{\text{low}}) \cdot (T_{LD}(w_{\text{low}}) + T_R(w_{\text{low}}))$ and with probability at least $1 - w_{\text{low}}^{O(1)}$ outputs a list $L$ of size $|L| \leqslant k + O(\varepsilon/w_{\text{low}})$ where, for each $i \in [k]$, there exists $\hat{\mu} \in L$ such that*

$$\|\hat{\mu} - \mu_i\| = O\left(\min_{1 \leqslant t' \leqslant t} \sqrt{t'}(1/\tilde{w}_i)^{1/t'} + f(\min(\tilde{w}_i, 1/3))\right).$$

*If the relative weight of the $i$-th cluster is large, i.e., $\tilde{\varepsilon}_i \leqslant 0.001$, then the error is further bounded by*

$$\|\hat{\mu} - \mu_i\| = O\left(g(\tilde{w}_i)\right).$$

The proof together with a more general statement, Theorem C.1, can be found in Appendix C.

Note that for a mixture setting with $k \geqslant 2$, the assumption $w_{\text{low}} \leqslant 1/k \leqslant 1/2$ is automatically fulfilled. Also, for large weights $\tilde{w}_i$ such that $\log(1/\tilde{w}_i) \ll t$, the $t'$ that minimizes $\sqrt{t'}(1/\tilde{w}_i)^{1/t'}$ is smaller than $t$, and for small weights the minimizer is $t' = t$.

**Gaussian case** For Gaussian inlier distributions, LD-ME and RME base learners with guarantees for Assumption 3.1 have already been developed in prior work. We can thus readily use them in the meta-algorithm to arrive at the following statement with the relative proportions defined in Eq. (3.1).

**Corollary 3.4** (Gaussian case). *Let $d, k, w_{\text{low}}$ and $t$ be as in Theorem 3.3. Let $\mathcal{X}$ be as in Eq. (2.1) with $D_i(\mu_i) = \mathcal{N}(\mu_i, I_d)$ with $\mu_i$'s satisfying $\|\mu_i - \mu_j\| \gtrsim \sqrt{\log 1/w_{\text{low}}}$ for all $i \neq j \in [k]$. There exists an algorithm that for $t = O(\log 1/w_{\text{low}})$, given $N = \text{poly}(d^t, (1/w_{\text{low}})^t)$ i.i.d. samples from $\mathcal{X}$ and $w_{\text{low}}$, runs in $\text{poly}(N)$ time and outputs a list $L$ such that with high probability $|L| = k + O(\varepsilon/w_{\text{low}})$ and, for all $i \in [k]$, there exists $\hat\mu \in L$ such that*

$$\|\hat\mu - \mu_i\| = O\left(\sqrt{\log 1/\tilde{w}_i}\right).$$

*If the relative weight of the $i$-th cluster is large, i.e. $\tilde{\varepsilon}_i \leqslant 0.001$, then the error is further bounded by*

$$\|\hat\mu - \mu_i\| = O\left(\tilde{\varepsilon}_i \sqrt{\log 1/\tilde{\varepsilon}_i}\right).$$

*Proof.* Theorem 6.12 from [5] provides an LD-ME algorithm $\mathcal{A}_{\text{kLD}}$ achieving error $f(\alpha) \leqslant O(\sqrt{t'}(1/\alpha)^{1/t'})$ for all $t' \leqslant t$. The sample and time complexity scale as $\text{poly}(d^t, (1/\alpha)^t)$. Also, Theorem 5.1 from [10] provides a robust mean estimation algorithm $\mathcal{A}_R$ such that for a small enough constant fraction of outliers $\varepsilon = 1 - \alpha$ it achieves error $g(\alpha) = O((1-\alpha)\sqrt{\log 1/(1-\alpha)})$ with sample complexity $\tilde{\Omega}(d/\varepsilon^2)$. Using these $\mathcal{A}_{\text{kLD}}$ and $\mathcal{A}_R$, we recover the desired bounds. $\square$

**Comparison with prior work** We now compare our result with the only previous method that can achieve guarantees in the LD-ML setting with unknown $w_i$. As discussed in [3], algorithms for the simple list-decoding model with $\alpha = w_{\text{low}}$ can be used for LD-ML by viewing a single mixture component as the "ground truth" distribution and effectively treating all other inlier components and original outliers as outliers. Besides requiring a much larger list size of $O(1/w_{\text{low}}) \gg k + O(\varepsilon/w_{\text{low}})$ and error $O(\sqrt{\log 1/w_{\text{low}}})$, this approach has two drawbacks that manifest in the suboptimal guarantees: 1) For larger clusters $i$ with $w_i \gg w_{\text{low}}$, LD-ME only achieves an error $O\left(\sqrt{\log 1/w_{\text{low}}}\right)$. Our result, even without separation assumption, achieves a sharper error bound $O\left(\sqrt{\log 1/w_i}\right)$. 2) When the mixture is separated, LD-ME cannot exploit the structure since it still models the data as $w_{\text{low}}\mathcal{N}(\mu_i, I_d) + (1 - w_{\text{low}})Q$ for each $i$, so that the algorithm inevitably treats all other true components as outliers. This results in the error $O\left(\sqrt{\log 1/w_{\text{low}}}\right) \gg O\left(\sqrt{\log 1/\tilde{w}_i}\right) = O(1)$ (when $\varepsilon \sim w_i \ll 1$). We refer to Appendix A for further illustrative examples. As a simple example, consider the uniform inlier mixture with $\varepsilon = w_i = 1/(k+1)$, where $k$ is large. In this case, previous results have error guarantees $O(\sqrt{\log k})$, while we obtain error $O(1)$.

**Separation assumption** For the problem of learning mixture models, a separation assumption is common in the literature [3, 9, 11, 12]. We require separation $\|\mu_i - \mu_j\| \gtrsim \sqrt{t}(1/w_{\text{low}})^{4/t}$, which we believe to be sub-optimal for the case of finite $t$. In cases when $\varepsilon$ is small (namely $\varepsilon \lesssim w_{\text{low}}$), there exist prior works on clustering allowing smaller separation. Specifically, when $t = 2$, a recent work [13] only requires $\|\mu_i - \mu_j\| \gtrsim 1/\sqrt{w_{\text{low}}}$. For a general $t \geqslant 4$, [9] succeeds under separation $(1/w_{\text{low}})^{2/t}$. We leave the possible relaxation of the separation requirement in the case of general $t$ and large $\varepsilon$ for the future work.

In the Gaussian case we require separation $\|\mu_i - \mu_j\| \gtrsim \sqrt{\log 1/w_{\text{low}}}$, which is optimal in the uniform ($w_i = 1/k$) case. Indeed, without the separation assumption, even in the *noiseless* uniform Gaussian case, [6] shows that no efficient algorithm can obtain error asymptotically better than $\Omega(\sqrt{\log 1/w_i})$. In Corollary B.5, we prove that the inner stage (Algorithm 3) of our algorithm, without knowledge of $w_i$ and separation assumption, achieves with high probability matching error guarantees $O(\sqrt{\log 1/w_i})$ with list size bounded by $O(1/w_{\text{low}})$.

### 3.2 Information-theoretical lower bounds and optimality

Next, we present information-theoretical lower bounds for list-decodable mixture learning on well-separated distributions $\mathcal{X}$ as defined in Eq. (2.1). We show that our error is optimal as long as the list size is required to be small. Our proof uses a simple reduction technique and leverages established lower bounds in [3] for the list-decodable mean estimation model (sLD in Section 2).

---

**Algorithm 1** Outer stage, informal (see Algorithm 6)

---

**Input:** $X$, $w_{\text{low}}$, $\Delta$, and sLD algorithm $\mathcal{A}_{\text{sLD}}$.
**Output:** Collection of sets $\mathcal{T}$.
1: $L \leftarrow (\hat{\mu}_1, \dots, \hat{\mu}_M) := \mathcal{A}_{\text{sLD}}(X)$ with $w_{\text{low}}$;
2: **while** $L \neq \emptyset$ **do**
3:      **for** $\hat{\mu} \in L$ **do**
4:          compute for *an appropriate distance function d*
$$S^{(1)}_{\hat{\mu}} = \{x \in X \mid d(x, \hat{\mu}) \leqslant \Delta\}, \quad S^{(2)}_{\hat{\mu}} = \{x \in X \mid d(x, \hat{\mu}) \leqslant 3\Delta\}$$
5:      **if** for all $\hat{\mu}$, $|S^{(2)}_{\hat{\mu}}| > 2|S^{(1)}_{\hat{\mu}}|$ **then** add $X$ to $\mathcal{T}$ and update $L \leftarrow \emptyset$
6:      **else**
7:          $\tilde{\mu} \leftarrow \operatorname{argmax}_{|S^{(2)}_{\hat{\mu}}| \leqslant 2|S^{(1)}_{\hat{\mu}}|} |S^{(1)}_{\hat{\mu}}|$
8:          add $S^{(2)}_{\tilde{\mu}}$ to $\mathcal{T}$
9:          $X \leftarrow X \setminus S^{(1)}_{\tilde{\mu}}$
10: **return** $\mathcal{T}$

---

**Proposition 3.5** (Information-theoretic lower bounds). *Let $\mathcal{A}$ be an algorithm that, given access to $\mathcal{X}$, outputs a list $L$ that, with probability $\geqslant 1/2$, for each $i \in [k]$ contains $\hat{\mu} \in L$ with $\|\hat{\mu} - \mu_i\| \leqslant \beta_i$.*

*(a) Consider the case with $\|\mu_i - \mu_j\| \gtrsim (1/w_{\text{low}})^{4/t}$ for $i \neq j \in [k]$, $D_i(\mu_i)$ having $t$-th bounded sub-Gaussian central moments and $\beta_i \leqslant C(1/w_{\text{low}})^{1/t}$ for each $i \in [k]$. If for some $s \in [k]$ it holds that $w_s \leqslant \varepsilon$, then algorithm $\mathcal{A}$ must either have error bound $\beta_s = \Omega((1/\tilde{w}_i)^{1/t})$ or $|L| \geqslant k + d - 1$.*

*(b) Consider the case with $\|\mu_i - \mu_j\| \gtrsim \sqrt{\log 1/w_{\text{low}}}$ for $i \neq j \in [k]$, $D_i(\mu_i) = \mathcal{N}(\mu_i, I_d)$ and $\beta_i \leqslant C\sqrt{\log 1/w_{\text{low}}}$ for each $i \in [k]$. If for some $s \in [k]$ it holds that $w_s \leqslant \varepsilon$, then algorithm $\mathcal{A}$ must either have error bound $\beta_s = \Omega(\sqrt{\log 1/\tilde{w}_i})$ or $|L| \geqslant k + \min\{2^{\Omega(d)}, (1/\tilde{w}_i)^{\omega(1)}\}$.*

In the Gaussian inlier case, Corollary 3.4 together with Proposition 3.5 imply optimality of our meta-algorithm. Indeed, if one plugs in optimal base learners (as in the proof of Corollary 3.4), we obtain error guarantee that matches lower bound. In particular, "exponentially" larger list size is necessary for asymptotically smaller error. For inlier components with bounded sub-Gaussian moments, [3] obtains information-theoretically (nearly-)optimal LD-ME base learners.

Furthermore, in [3], formal evidence of computational hardness was obtained (see their Theorem 5.7, which gives a lower bound in the statistical query model introduced by [14]) that suggests obtaining error $\Omega_t((1/\tilde{w}_s)^{1/t})$ requires running time at least $d^{\Omega(t)}$. This was proved for Gaussian inliers and the running time matches ours up to a constant in the exponent.

## 4 Algorithm sketch

We now sketch our meta-algorithm specialized to the case of separated Gaussian components $\mathcal{N}(\mu_i, I_d)$ and provide intuition for how it achieves the guarantees in Corollary 3.4. In this section, we only discuss how to obtain an error of $O(\sqrt{\log 1/\tilde{w}_i})$ for each mean when $\varepsilon \gtrsim \min_i w_i$. We refer to Appendix D for how to achieve the refined error guarantee of $O(\tilde{\varepsilon}_i \sqrt{\log 1/\tilde{\varepsilon}_i})$ when $\tilde{\varepsilon}_i$ is small.

As discussed in Section 3.1, running an out-of-the-box LD-ME algorithm for the sLD problem on our input with parameter $\alpha = w_{\text{low}}$ would give sub-optimal guarantees. In contrast, our two-stage Algorithm 2, equipped with the appropriate cor-kLD and RME base learners as depicted in Figure 1, obtains for each component an error guarantee that is as good as if we had access to the samples *only* from this component and from the outliers. We now give more details about the outer stage, Algorithm 1, and inner stage, Algorithm 3, and describe on a high-level how they contribute to a short output list with optimal error bound in Corollary 3.4 for large outlier fractions.

### 4.1 Inner stage: list-decodable mean estimation with unknown inlier fraction

We now describe how to use a black-box cor-kLD algorithm to obtain a list-decoding algorithm $\mathcal{A}_{\text{aLD}}$ for the cor-aLD mean-estimation setting with access only to $\alpha_{\text{low}} \leqslant \alpha$. $\mathcal{A}_{\text{aLD}}$ is used

in the proof of Corollary B.5 and plays a crucial role (see Figure 1) in our meta-algorithm. In particular, it deals with the unknown weight of the inlier distribution in each set returned by the outer stage. Note that estimating $\alpha$ from the input samples is impossible by nature. Indeed, we cannot distinguish between potential outlier clusters of arbitrary proportion $\leqslant 1 - \alpha$ and the inlier component. Underestimating the size of a large component would inevitably lead to a suboptimal error guarantee. We now show how to overcome this challenge and achieve an error guarantee $O(\sqrt{\log 1/\alpha})$ for a list size $1 + O((1 - \alpha)/\alpha_{\mathrm{low}})$ for the cor-aLD setting. Here we only outline our algorithm and refer to Appendix D for the details.

Algorithm 3 first produces a large list of estimates corresponding to many potential values of $\alpha$ and then prunes it while maintaining a good estimate in the list. In particular, for each $\hat{\alpha} \in A := \{\alpha_{\mathrm{low}}, 2\alpha_{\mathrm{low}}, \ldots, \lfloor 1/(3\alpha_{\mathrm{low}}) \rfloor \alpha_{\mathrm{low}}\}$, we run $\mathcal{A}_{\mathrm{kLD}}$ with parameter $\hat{\alpha}$ to obtain a list of means. We append $\hat{\alpha}$ to each mean in the list and obtain a list of pairs $(\hat{\mu}, \hat{\alpha})$. We concatenate these lists of pairs for all $\hat{\alpha}$ and obtain a list $L$ of size $O(1/\alpha_{\mathrm{low}}^2)$. By design, one element of $A$ is close to the true $\alpha$, so the list $L$ contains at least one $\hat{\mu}$ that is $O(\sqrt{\log 1/\alpha})$-close — the error guarantee that we aim for — and there is indeed at least an $\alpha$-fraction of samples near $\hat{\mu}$. We call such a hypothesis "nearby".

Finally, we prune this concatenated list by verifying for each $\hat{\mu}$ whether there is indeed an $\hat{\alpha}$-fraction of samples "not too far" from it. This is similar to pruning procedures with known $\alpha$ proposed in prior work (see Proposition B.1 in [3]). Our procedure (i) never discards a "nearby" hypothesis, and outputs a list where (ii) every hypothesis contains a sufficient number of points close to it and (iii) all hypotheses are separated. Property (i) implies that the final error is $O(\sqrt{\log 1/\alpha})$ and properties (ii) and (iii) imply list size bound $1 + O((1 - \alpha)/\alpha_{\mathrm{low}})$. Note that when $\alpha < \alpha_{\mathrm{low}}$, the list size can be simply upper bounded by $O(1/\alpha_{\mathrm{low}})$, see Remark B.4.

### 4.2 Two-stage meta-algorithm

Even though we could run $\mathcal{A}_{\mathrm{aLD}}$ on the entire dataset with $\alpha_{\mathrm{low}} = w_{\mathrm{low}}$, we would only achieve an error for the $i^{\mathrm{th}}$ inlier cluster mean of $O(\sqrt{\log 1/w_i})$ – which can be much larger than $O(\sqrt{\log 1/\tilde{w}_i})$ – for a list of size $O(1/w_{\mathrm{low}})$. While $\mathcal{A}_{\mathrm{aLD}}$ takes into account the unknown weight of the clusters, it still treats other inlier clusters as outliers. We now show that if the outer stage Algorithm 1 of our meta-algorithm Algorithm 2 separates the samples into a not-too-large collection $\mathcal{T}$ of sets with certain properties, running $\mathcal{A}_{\mathrm{aLD}}$ separately on each of the sets can lead to the desired guarantees. In particular, let us assume that $\mathcal{T}$ consists of potentially overlapping sets such that:

(1) For each inlier cluster $C^*$, there exists one set $T \in \mathcal{T}$ such that $T$ contains (almost) all points from $C^*$ and at most $O(\varepsilon n)$ other points,

(2) It holds that $\sum_{T \in \mathcal{T}} |T| \leqslant n + O(\varepsilon n)$.

By (1), for every inlier cluster $C^*$ with a corresponding true weight $w^*$, there exists a set $T$ such that the points from $C^*$ constitute at least an $\tilde{w}$-fraction of $T$ with $\tilde{w} := \Omega(w^*/(w^* + \varepsilon))$. By Section 4.1, applying $\mathcal{A}_{\mathrm{aLD}}$ with $\alpha_{\mathrm{low}} = w_{\mathrm{low}} \cdot n/|T|$ on such a $T$ then yields a list of size $1 + O((1 - \tilde{w})/w_{\mathrm{low}})$ with an estimation error at most $O(\sqrt{\log 1/\tilde{w}})$. If $T$ contains (almost) no inliers, that is, there is no inlier component that should recovered, then $\mathcal{A}_{\mathrm{kLD}}$ returns a list of size $O(|T|/(w_{\mathrm{low}}n))$.

Now, by the two properties, (almost) all inlier points lie in at most $k$ sets of $\mathcal{T}$, and all other sets of $\mathcal{T}$ contain in total at most $O(\varepsilon n)$ points. Hence, concatenating all lists outputted by $\mathcal{A}_{\mathrm{aLD}}$ applied to all $T \in \mathcal{T}$ leads to a final list size bounded by $k + O(\varepsilon/w_{\mathrm{low}})$.

### 4.3 Outer stage: separating inlier clusters

We now informally describe the outer stage that produces the collection of sets $\mathcal{T}$ with the desiderata described in Section 4.2, leaving the details to Appendix E. The main steps are outlined in pseudocode in Algorithm 1.

Given a set $X$ of $N = \mathrm{poly}(d^t, 1/w_{\mathrm{low}})$ i.i.d. input samples from the distribution Eq. (2.1) with Gaussian inlier components, the first step of the meta-algorithm is to run Algorithm 1 on $X$ and $w_{\mathrm{low}}$ with $\Delta = O(\sqrt{\log 1/w_{\mathrm{low}}})$. Algorithm 1 runs an sLD algorithm on the samples and produces a (large) list of estimates $L$ such that, for each mean, at least one estimate is $O(\sqrt{\log 1/w_{\mathrm{low}}})$-close to it. It then add sets to $\mathcal{T}$ that correspond to these estimates via a dynamic "two-scale" process.

Specifically, for each $\hat{\mu} \in L$, we construct *two sets* $S_{\hat{\mu}}^{(1)} \subseteq S_{\hat{\mu}}^{(2)}$ consisting of samples close to $\hat{\mu}$. By construction, we guarantee that if $S_{\hat{\mu}}^{(1)}$ contains a non-negligible fraction of samples from any inlier cluster $C^*$, then $S_{\hat{\mu}}^{(2)}$ contains (almost) all samples from $C^*$ (see Theorem B.7 (ii)).

Now we very briefly illustrate how this process could be helpful in proving properties (1) and (2). Observe that, as long as there exists some $\hat{\mu}$ with $|S_{\hat{\mu}}^{(2)}| \leqslant 2|S_{\hat{\mu}}^{(1)}|$, we add $S_{\hat{\mu}}^{(2)}$ to $\mathcal{T}$ and remove the samples from $S_{\hat{\mu}}^{(1)}$. Consider one such $\hat{\mu}$. For property (1), we merely note that if $S_{\hat{\mu}}^{(1)}$ contains a part of an inlier cluster $C^*$, then $S_{\hat{\mu}}^{(2)}$ contains (almost) all of $C^*$, so we add to $\mathcal{T}$ a set that contains (almost) all of $C^*$; otherwise, when we remove $S_{\hat{\mu}}^{(1)}$ we remove (almost) no points from $C^*$, so (almost) all the points from $C^*$ remain in play. For property (2), we merely note that whenever we add $S_{\hat{\mu}}^{(2)}$ to $\mathcal{T}$, increasing the number of points in it by $|S_{\hat{\mu}}^{(2)}|$, we also remove the samples from $S_{\hat{\mu}}^{(1)}$, reducing the number of samples by $|S_{\hat{\mu}}^{(1)}| \geqslant |S_{\hat{\mu}}^{(2)}|/2$. The proof of the properties uses some additional arguments of a similar flavor, and we defer it to Appendix E.

## 5  Related work

**List-decodable mean estimation**  Inspired by the list-decoding paradigm that was first introduced for error-correcting codes for large error rates [15], list-decodable mean estimation has become a popular approach for robustly learning the mean of a distribution when the majority of the samples are outliers. A long line of work has proposed efficient algorithms with theoretical guarantees. These algorithms are either based on convex optimization [9, 16], a filtering approach [3, 17], or low-dimensional projections [18]. Near-linear time algorithms were obtained in [19] and [8]. The list-decoding paradigm is not only used for mean estimation but also other statistical inference problems. Examples include sparse mean estimation [20, 21], linear regression [22–24], subspace recovery [25, 26], clustering [27], stochastic block models and crowd sourcing [16, 28].

**Robust mean estimation and mixture learning**  When the outliers constitute a minority, algorithms typically achieve significantly better error guarantees than in the list-decodable setting. Robust mean estimation algorithms output a single vector close to the mean of the inliers. In a variety of corruption models, efficient algorithms are known to achieve (nearly) optimal error

Robust mixture learning tackles the model in Eq. (2.1) with $\varepsilon \ll \min_i w_i$ and aims to output exactly $k$ vectors with an accurate estimate for the population mean of each component [3, 4, 7, 9, 11, 13, 29, 30]. These algorithms do not enjoy error guarantees for clusters with weights $w_i < \varepsilon$. To the best of our knowledge, our algorithm is the first to achieve non-trivial guarantees in this larger noise regime.

**Robust clustering**  Robust clustering [31] also addresses the presence of small fractions of outliers in a similar spirit to robust mixture learning, conceptually implemented in the celebrated DBScan algorithm [32]. Assuming the output list size is large enough to capture possible outlier clusters, these methods may also be used to tackle list-decodable mixture learning - however, they do not come with an inherent procedure to determine the right choice of hyperparameters that ultimately output a list size that adapts to the problem.

## 6  Discussion and future work

In this work, we prove that even when small groups are outnumbered by adversarial data points, efficient list-decodable algorithms can provide an accurate estimation of all means with minimal list size. The proof for the upper bound is constructive and analyzes a plug-and-play meta-algorithm (cf. Figure 1) that inherits guarantees of the black-box cor-kLD algorithm $\mathcal{A}_{\mathrm{kLD}}$ and RME algorithm $\mathcal{A}_R$, which it uses as base learners. Notably, when the inlier mixture is a mixture of Gaussians with identity covariance, we achieve optimality. Furthermore, any new development for the base learners automatically translates to improvements in our bounds.

We would like to end by discussing the possible practical impact of this result. Since an extensive empirical study is out of the scope of this paper, besides the fact that ground-truth means for unsupervised real-world data are hard to come by, we provide preliminary experiments on synthetic data. Specifically, we generate data from a separated $k-$Gaussian mixture with additive contaminations as in Eq. (2.1) and different types of adversarial distributions (see detailed description in Appendix I). We focus on the regime $\varepsilon \sim w_i$ where our algorithms shows the largest theoretical improvements.

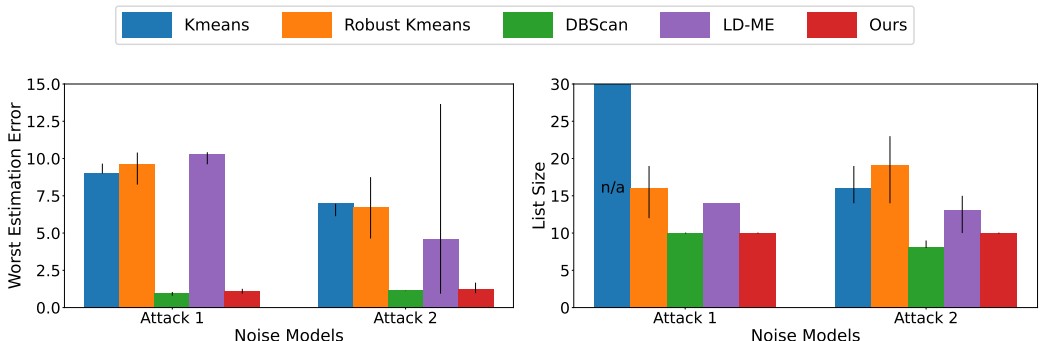

Figure 2: Comparison of five algorithms with two adversarial noise models. The attack distributions and further experimental details are given in Appendix I. On the left we show worst estimation error for constrained list size and on the right the smallest list size for constrained error guarantee. We plot the median of the metrics with the error bars showing 25th and 75th percentile.

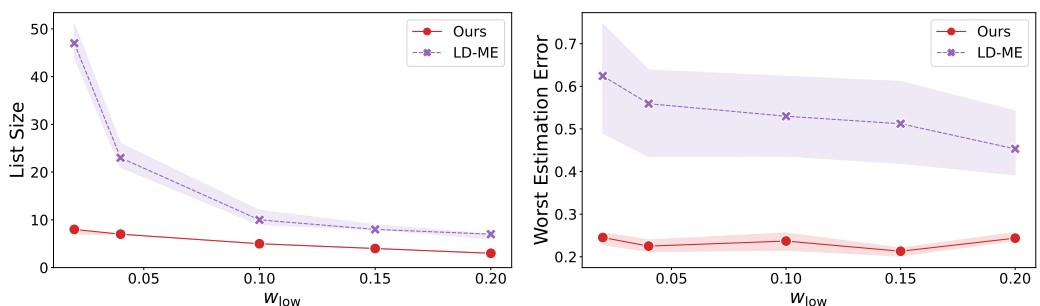

Figure 3: Comparison of list size and estimation error for large inlier cluster for varying $w_{\text{low}}$ inputs. The experimental setup is illustrated in Appendix I. We plot the median values with error bars showing 25th and 75th quantiles. As $w_{\text{low}}$ decreases, we observe a roughly constant estimation error for our algorithm while the error for LD-ME increases. Further, the decrease in list size is much more severe for LD-ME than for our algorithm.

We then compare the output of our algorithm with the vanilla LD-ME algorithm from [8] with $w_{\text{low}} = 0.02$ and (suboptimal) LD-ML guarantees as well as well-known (robust) clustering heuristics without LD-ML guarantees, such as the $k$-means [33], Robust $k$-means [34], and DBSCAN [32]. Even though none of these heuristics have LD-ML guarantees, they are commonly used and known to also perform well in practice in noisy settings. In Figure 2 (left), we fix the list size to 10 and plot the errors for the worst inlier cluster, typically the smallest. We compare the performance of the algorithms by plotting the worst-case estimation errors for a given list size and list sizes that algorithms require to achieve a given worst-case estimation error. In Figure 2 (right), we fix the error and plot the minimal list size at which competing algorithms reach the same or smaller worst estimation error. Further details on the experiments are provided in Appendix I. In a different experiment (see Figure 3 and Appendix I.1 for details), we observe that our approach outperforms LD-ME when $w_{\text{low}}$ varies, both in achieving smaller list size and smaller estimation error.

Overall, in line with our theory, our method significantly outperforms the LD-ME algorithm, and performs better or on par with the heuristic approaches. Additional experimental comparison and implementation details can be found in Appendix I. Even though these experiments do not allow conclusive statements about the improvement of our algorithm for mixture learning for real-world data, they do provide encouraging evidence that effort could be well-spent on follow-up empirical and theoretical work building on our results. For example, it would be interesting to conduct a more extensive empirical study comparing our algorithm with a variety of robust clustering algorithms. Additionally, practical data often contains components with varying scales. An interesting direction for future work could be to extend our algorithm to handle differently scaled covariances in an agnostic manner.

## Acknowledgements

DD is supported by ETH AI Center doctoral fellowship and ETH Foundations of Data Science initiative. RB, ST, GN, and DS have received funding from the European Research Council (ERC) under the European Union's Horizon 2020 research and innovation programme (grant agreement No 815464).

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

# A Examples

$k$ **inlier cluster,** $c$ **outlier clusters.** One tricky adversarial distribution is the Gaussian mixture model itself. In particular, we consider

$$\mathcal{X}_c = \frac{k}{k+c} \sum_{i=1}^{k} \frac{1}{k} \mathcal{N}(\mu_i, I) + \frac{c}{k+c} \sum_{i=1}^{c} \frac{1}{c} \mathcal{N}(\tilde{\mu}_i, I), \tag{A.1}$$

where the first $k$ Gaussian components are inliers and $Q$ is a GMM with $c$ components, which we call *fake* clusters. Since all inlier cluster weights are identical, we denote $w \coloneqq w_i = 1/(k+c)$. Assume that $1 \ll c \ll k$, which corresponds to $\varepsilon \gg w$. Then, relative weights are $\tilde{w} = 1/(c+1) \approx 1/c$. Due to large adversary, previous results on learning GMMs cannot be applied, leaving vanilla list-decodable learning. However, the latter also cannot guarantee anything better that $\Omega(\sqrt{t}k^{1/t})$ even with the knowledge of $k$, as long as list size is $O(k+c)$, which can be much worse than our guarantees of $O(\sqrt{t}c^{1/t})$ for the same list size.

Their drawback is that they do not utilize separation between true clusters, i.e., for each $i$, they model the data as

$$\mathcal{X} = \frac{1}{k+c} \mathcal{N}(\mu_i, I) + \left(1 - \frac{1}{k+c}\right) Q.$$

where $Q$ can be "arbitratily adversarial" for recovering $\mu_i$.

**Big + small inlier clusters** Consider the mixture

$$\mathcal{X}_b = (1 - w - \varepsilon)\,\mathcal{N}(\mu_1, I_d) + w\mathcal{N}(\mu_2, I_d) + \varepsilon Q, \tag{A.2}$$

where $\|\mu_1 - \mu_2\| = \Omega(\sqrt{\log 1/w})$, $w \ll \varepsilon \ll 1$, and $Q$ is chosen adversarially. In this example we have two inlier clusters, one with large weight $\approx 1$ and another with small weight $w$. Adversarial distribution $Q$ has large weight relative to the small cluster, but still negligible weight compared to the large one.

Previous methods would either (i) recover large cluster with optimal error $O(\varepsilon)$ (see, e.g., [35]) but miss out small cluster or (ii) recover both clusters using list-decodable mean estimation with known $\alpha = w$, but with suboptimal errors $O(\sqrt{\log 1/w})$ and list size $O(1/w)$. In contrast, Corollary 3.4 guarantees list size at most $1 + O(\varepsilon/w)$, error $O(\sqrt{\log \varepsilon/w})$ for the small cluster, and error $O(\varepsilon\sqrt{\log 1/\varepsilon})$ for the larger. In general, we achieve (i) optimal errors for both clusters and (ii) optimal (up to constants) list size.

# B Inner and outer stage algorithms and guarantees

Our meta-algorithm Algorithm 2 assumes black-box access to a list-decodable mean estimation algorithm and a robust mean estimation algorithm for sub-Gaussian (up to the $t^{\text{th}}$ moment) distributions. From these we obtain stronger mean estimation algorithms when the fraction of outliers is unknown, and finally stronger algorithms for learning separated mixtures when the fraction of outliers can be arbitrarily large. Our algorithm achieves guarantees with polynomial runtime and sample complexity if the black-box learners achieve the guarantees for their corresponding mean estimation setting. In this section we discuss the corruption model and inner and out stage of the meta-algorithm in detail and prove properties needed for the proof of the main Theorem 3.3.

## B.1 Detailed setting

In order to achieve these guarantees, our black-box algorithms need to work under a model in which an adversary is allowed to remove a small fraction of the inliers and to add arbitrarily many outliers. In our proofs, for simplicity of exposition, we require the algorithms to have mean estimation guarantees for a small adversarially removed fraction of $w_{\text{low}}^2$. Formally, the corruption model as defined as follows.

**Definition B.1** (Corruption model). Let $d \in \mathbb{N}_+$, and $\alpha \in [w_{\text{low}}, 1]$. Let $D$ be a $d$-dimensional distribution. An input of size $n$ according to our corruption model is generated as follows:

**Algorithm 2** FullAlgorithm

---

**Input:** Samples $S = \{x_1, \ldots, x_n\}$, $w_{\text{low}}$, algorithms $\mathcal{A}_{\text{kLD}}$, and $\mathcal{A}_R$.
**Output:** List $L$.
 1: Run OuterStage (Algorithm 6) on $S$ and let $\mathcal{T}$ be the returned list.
 2: $L \leftarrow \emptyset$.
 3: **for** $T \in \mathcal{T}$ **do**
 4:     Run InnerStage (Algorithm 3) on $T$ with $\alpha_{\text{low}} = w_{\text{low}} \cdot \frac{n}{|T|}$.
 5:     Add the elements of the returned list to $L$.
 6: **return** $L$.

---

**Algorithm 3** InnerStage

---

**Input:** Samples $S = \{x_1, \ldots, x_n\}$, $\alpha_{\text{low}} \in [w_{\text{low}}, 1]$, $\mathcal{A}_{\text{kLD}}$, and $\mathcal{A}_R$.
**Output:** List $L$.
 1: $\alpha_{\text{low}} \leftarrow \min(1/100, \alpha_{\text{low}})$
 2: $M \leftarrow \emptyset$
 3: **for** $\hat{\alpha} \in \{\alpha_{\text{low}}, 2\alpha_{\text{low}}, \ldots, \lfloor 1/(3\alpha_{\text{low}}) \rfloor \alpha_{\text{low}}\}$ **do**
 4:     run $\mathcal{A}_{\text{kLD}}$ on $S$ with fraction of inliers set to $\hat{\alpha}$
 5:     add the pair $(\hat{\mu}, \hat{\alpha})$ to $M$ for each output $\hat{\mu}$
 6: Let $L$ be the output of ListFilter (Algorithm 4) run on $S$, $\alpha_{\text{low}}$, and $M$
 7: **for** $(\hat{\mu}, \hat{\alpha}) \in L$ **do**
 8:     replace $\hat{\mu}$ by the output of ImproveWithRME (Algorithm 5) run on $S$, $\hat{\mu}$, $\tau = 40\psi_t(\hat{\alpha}) + 4f(\hat{\alpha})$, and $\mathcal{A}_R$
 9: **return** $L$

---

- Draw a set $C^*$ of $n_1 = \lceil \alpha n \rceil$ i.i.d. samples from the distribution $D$.

- An adversary is allowed to arbitrarily remove $\lfloor w_{\text{low}}^2 n_1 \rfloor$ samples from $C^*$. We refer to the resulting set as $S^*$ with size $n_2 = |S^*|$.

- An adversary is allowed to add $n - n_2$ arbitrary points to $S^*$. We refer to the resulting set as $S_{\text{adv}}$ with size $n_3 = |S_{\text{adv}}|$.

- If $n_3 < n$, pad $S_{\text{adv}}$ with $n - n_3$ arbitrary points and call the resulting set $S$.

- Return $S$.

We call cor-kLD the model when $w_{\text{low}}$ and $\alpha$ are given to the algorithm and cor-aLD the model when $w_{\text{low}}$ and lower bound $\alpha_{\text{low}} \geqslant w_{\text{low}}$ are given to the algorithm, such that $\alpha \geqslant \alpha_{\text{low}}$. Note that $\alpha$ is **not** provided in cor-aLD model.

Note that in Definition B.1 $|S| = n$ and $S^*$ constitutes at least an $\alpha(1 - w_{\text{low}}^2)$-fraction of $S$.

## B.2 Inner stage algorithm and guarantees

The algorithm consists of three steps: (1) Constructing a list of hypotheses, (2) Filtering the hypotheses, and (3) Improving the hypotheses if $\alpha \geqslant 1 - \varepsilon_{\text{RME}}$. For convenience, we restate the InnerStage algorithm introduced in the main text.

**Theorem B.2** (Inner stage guarantees). *Let $d \in \mathbb{N}_+$, $w_{\text{low}} \in (0, 10^{-4}]$, $w_{\text{low}} \leqslant \alpha_{\text{low}} \leqslant \alpha \leqslant 1$, and $t$ be an even integer. Let $D(\mu^*)$ be a $d$-dimensional distribution with mean $\mu^* \in \mathbb{R}^d$ and sub-Gaussian $t$-th central moments.*

*Consider the* cor-aLD *corruption model in Definition B.1 with parameters $d$, $w_{\text{low}}$, $\alpha$ and distribution $D = D(\mu^*)$. Let $\mathcal{A}_{\text{kLD}}$ and $\mathcal{A}_R$ satisfy Assumption 3.1 with high success probability (see Remark B.3).*

*Then* InnerStage *(Algorithm 3), given an input of* $\text{poly}(d, 1/w_{\text{low}}) \cdot (N_{LD}(w_{\text{low}}) + N_R(w_{\text{low}}))$ *samples from the* cor-aLD *corruption model, and access to the parameters $d$, $w_{\text{low}}$, $\alpha_{\text{low}}$,*

---
**Algorithm 4** ListFilter
---
**Input:** Samples $S = \{x_1, \ldots, x_n\}$, $\alpha_{\text{low}} \in [w_{\text{low}}, 1/100]$, and $M = \{(\hat{\mu}_1, \hat{\alpha}_1), \ldots, (\hat{\mu}_m, \hat{\alpha}_m)\}$
**Output:** List $L$
1: define $\beta(\alpha) = 10\psi_t(\alpha) + f(\alpha)$
2: let $v_{ij}$ be a unit vector in the direction of $\hat{\mu}_i - \hat{\mu}_j$ for $\hat{\mu}_i \neq \hat{\mu}_j \in \{\hat{\mu}, \text{ for } (\hat{\mu}, \hat{\alpha}) \in M\}$
3: $J \leftarrow \emptyset$
4: **for** $(\hat{\mu}_i, \hat{\alpha}_i) \in M$ in decreasing order of $\hat{\alpha}_i$ **do**
5:     **if** exists $j \in J$, such that $\|\hat{\mu}_i - \hat{\mu}_j\| \leqslant 4\beta(\hat{\alpha}_i)$ **then continue**
6:     $T_i \leftarrow \bigcap_{j \in J}\{x \in S, \text{ s.t. } |v_{ij}^\top(x - \hat{\mu}_i)| \leqslant \beta(\hat{\alpha}_i)\}$.
7:     **if** $|T_i| < 0.9\hat{\alpha}_i n$ **then** remove $(\hat{\mu}_i, \hat{\alpha}_i)$ from $M$ and **continue**
8:     add $i$ to $J$
9:     **for** $j \in J \setminus \{i\}$ **do**
10:         $T_j \leftarrow T_j \bigcap \{x \in S, \text{ s.t. } |v_{ij}^\top(x - \hat{\mu}_j)| \leqslant \beta(\hat{\alpha}_i)\}$
11:         **if** $|T_j| < 0.9\hat{\alpha}_j n$ **then:**
12:             remove $(\hat{\mu}_j, \hat{\alpha}_j)$ from $M$
13:             rerun ListFilter (Algorithm 4) with the new $M$
14: **return** $\{(\hat{\mu}_i, \hat{\alpha}_i), \text{ for } i \in J\}$

---
**Algorithm 5** ImproveWithRME
---
**Input:** Samples $S = \{x_1, \ldots, x_n\}$, vector $\hat{\mu}$, threshold $\tau$, and $\mathcal{A}_R$
**Output:** A vector $\tilde{\mu} \in \mathbb{R}^d$
1: $\tilde{\beta} \leftarrow \tau$
2: let $\tilde{\alpha}$ be the smallest value in $[1 - \varepsilon_{\text{RME}}, 1]$ that satisfies $g(\tilde{\alpha}) \leqslant \tilde{\beta}/2$. If none exists, **return** $\hat{\mu}$
3: $\tilde{\mu} \leftarrow \hat{\mu}$ and let $\mu_{\text{RME}}$ be the output of $\mathcal{A}_R$ run on $S$ with inlier fraction set to $\tilde{\alpha}$.
4: **while** $\|\tilde{\mu} - \mu_{\text{RME}}\| \leqslant 3\tilde{\beta}/2$ **do**
5:     $\tilde{\mu} \leftarrow \mu_{\text{RME}}$
6:     $\tilde{\beta} \leftarrow g(\tilde{\alpha})$
7:     let $\tilde{\alpha}'$ be the smallest in $[\tilde{\alpha} + w_{\text{low}}^2, 1]$ such that $g(\tilde{\alpha}') \leqslant \tilde{\beta}/2$. If none exists, **break**
8:     $\tilde{\alpha} \leftarrow \tilde{\alpha}'$
9:     let $\mu_{\text{RME}}$ be the output of $\mathcal{A}_R$ on $S$ with inlier fraction set to $\tilde{\alpha}$
10: **return** $\tilde{\mu}$

---

and $t$, runs in time $\text{poly}(d, 1/w_{\text{low}}) \cdot (T_{LD}(w_{\text{low}}) + T_R(w_{\text{low}}))$ and outputs a list $L$ of size $|L| \leqslant 1 + O((1 - \alpha)/\alpha_{\text{low}})$ such that, with probability $1 - w_{\text{low}}^{O(1)}$,

    *1. There exists $\hat{\mu} \in L$ such that*

$$\|\hat{\mu} - \mu^*\| \leqslant O(\psi_t(\alpha/4) + f(\alpha/4)).$$

    *2. If $\alpha \geqslant 1 - \varepsilon_{\text{RME}}$, then there exists $\hat{\mu} \in L$ such that*

$$\|\hat{\mu} - \mu^*\| \leqslant O(g(\alpha - w_{\text{low}}^2)).$$

Proof of Theorem B.2 can be found in Appendix D.

**Remark B.3.** *For any $r \in \mathbb{N}$, we can increase probabilities of success of $\mathcal{A}_{\text{kLD}}$ and $\mathcal{A}_R$ from $1/2$ to $1 - 2^{-r}$ in the following way: we increase number of samples by a factor of $r$, randomly split $S$ into $r$ subsets of equal size, apply $\mathcal{A}_{\text{kLD}}$ and $\mathcal{A}_R$ to these subsets and concatenate their outputs. In the proofs we assume that the success probabilities are $1 - w_{\text{low}}^C$ for large enough constant $C$. This increases the size of the list returned by $\mathcal{A}_{\text{kLD}}$, the number of samples, and the running time by a factor $O(\log(1/w_{\text{low}}))$. In particular, we assume that the size of the list returned by $\mathcal{A}_{\text{kLD}}$ is much smaller than the inverse failure probability.*

**Remark B.4.** *In the execution of the meta-algorithm, it may happen that Algorithm 3 is run on set $T$ with almost no inliers, i.e., $\alpha < \alpha_{\text{low}}$. We note that from the analysis (see Appendix D, or [3], Proposition B.1), we always have upper bound $|L| = O(1/\alpha_{\text{low}})$.*

An immediate consequence of Theorem B.2 are the following guarantees of directly applying Algorithm 3 to the mixture learning case with no separation. Here we present upper bounds for Algorithm 3, when no separation assumptions are imposed.

**Corollary B.5.** *Let $d, k \in \mathbb{N}_+$, $w_{\mathrm{low}} \in (0, 10^{-4}]$, and $t$ be an even integer. For all $i = 1, \ldots, k$, let $D_i(\mu_i)$ be a $d$-dimensional distribution with mean $\mu_i \in \mathbb{R}^d$ and sub-Gaussian $t$-th central moments. Let $\varepsilon > 0$ and, for all $i = 1, \ldots, k$, let $w_i \in [w_{\mathrm{low}}, 1]$, such that $\sum_{i=1}^{k} w_i + \varepsilon = 1$. Let $\mathfrak{X}$ be the $d$-dimensional mixture distribution*

$$\mathfrak{X} = \sum_{i=1}^{k} w_i D_i(\mu_i) + \varepsilon Q,$$

*where $Q$ is an unknown adversarial distribution that can depend on all the other parameters. Let $\mathcal{A}_{\mathrm{kLD}}$ and $\mathcal{A}_R$ satisfy Assumption 3.1.*

*Then there exists an algorithm that, given $\mathrm{poly}(d, 1/w_{\mathrm{low}}) \cdot (N_{LD}(w_{\mathrm{low}}) + N_R(w_{\mathrm{low}}))$ i.i.d. samples from $\mathfrak{X}$, and given also $d$, $k$, $w_{\mathrm{low}}$, and $t$, runs in time $\mathrm{poly}(d, 1/w_{\mathrm{low}}) \cdot (T_{LD}(w_{\mathrm{low}}) + T_R(w_{\mathrm{low}}))$ and outputs a list $L$ of size $|L| = O(1/w_{\mathrm{low}})$, such that, with probability at least $1 - w_{\mathrm{low}}^{O(1)}$:*

1. *For each $i \in [k]$, there exists $\hat{\mu} \in L$ such that*

$$\|\hat{\mu} - \mu_i\| \leqslant O(\psi_t(w_i/4) + f(w_i/4)).$$

2. *For each $i \in [k]$, if $w_i \geqslant 1 - \varepsilon_{\mathrm{RME}}$, then there exists $\hat{\mu} \in L$ such that*

$$\|\hat{\mu} - \mu_i\| \leqslant O(g(w_i - w_{\mathrm{low}}^2)).$$

*Proof.* Proof follows by applying Theorem B.2 to $\mathfrak{X}$ with $\alpha_{\mathrm{low}} = w_{\mathrm{low}}$ and treating each component as a corresponding inlier distribution with $\alpha = w_i$. This gives error upper bound for all inlier components, furthermore, since $\alpha_{\mathrm{low}} = w_{\mathrm{low}}$, list size can be bounded as $|L| \leqslant 1 + O((1 - \alpha)/\alpha_{\mathrm{low}}) = O(1/w_{\mathrm{low}})$. $\square$

### B.3 Outer stage algorithm and guarantees

In the outer stage, presented in Algorithm 6, we make use of the list-decodable mean estimation algorithm in Theorem B.2 in order to solve list-decodable mixture estimation with separated means. We now present results on the outer stage algorithm. For ease of notation, when it's clear from the context, we drop the indices and refer to elements $\mu_j \in M$ for some $j \in [|M|]$ as $\mu$ and their corresponding sets $S_j^{(1)}, S_j^{(2)}$, as defined in lines 6–7 in Algorithm 6, as $S^{(1)}, S^{(2)}$. Further, for $i \in [k]$, let $C_i^*$ denote the set of points corresponding to the $i$-th inlier component, also called the $i$-th inlier cluster.

**Theorem B.6** (Outer stage guarantees, beginning of execution). *Let $S$ consist of $n$ i.i.d. samples from $\mathfrak{X}$ as in the statement of Theorem C.1. Run $\mathrm{OuterStage}$ (Algorithm 6) on $S$ and consider the first iteration of the while-loop and for each $\mu \in M$, denote the corresponding sets as $S^{(1)}, S^{(2)}$. Then, with probability at least $1 - w_{\mathrm{low}}^{O(1)}$, we have that*

(i) *the list $M$ that $\mathcal{A}_{\mathrm{sLD}}$ outputs has size $|M| \leqslant 2/w_{\mathrm{low}}$,*

(ii) *for each $i \in [k]$, there exists $m_i \in [|M|]$ such that $\left|S_{m_i}^{(1)} \cap C_i^*\right| \geqslant (1 - \frac{w_{\mathrm{low}}^2}{2}) |C_i^*|$,*

(iii) *for each $i \in [k]$ and $\mu \in M$, we have $\left|S^{(1)} \cap C_i^*\right| < w_{\mathrm{low}}^4 |C_i^*|$ or $\left|S^{(2)} \cap C_i^*\right| \geqslant (1 - \frac{w_{\mathrm{low}}^2}{2}) |C_i^*|$,*

(iv) *for each $i \in [k]$ and $\mu \in M$ such that $\left|S^{(2)} \cap C_i^*\right| \geqslant w_{\mathrm{low}}^4 |C_i^*|$, we have $\sum_{i' \in [k] \setminus \{i\}} \left|S^{(2)} \cap C_{i'}^*\right| \leqslant w_{\mathrm{low}}^4 n$,*

(v) *for $i \neq i' \in [k]$ and for $j, j' \in [|M|]$, if $|S_j^{(2)} \cap C_i^*| \geqslant w_{\mathrm{low}}^4 |C_i^*|$ and $|S_{j'}^{(2)} \cap C_{i'}^*| \geqslant w_{\mathrm{low}}^4 |C_{i'}^*|$, then $S_j^{(2)} \cap S_{j'}^{(2)} = \emptyset$.*

---

**Algorithm 6** OuterStage

---

**Input:** Samples $S = \{x_1, \ldots, x_n\}$, $w_{\text{low}}$, $\mathcal{A}_{\text{sLD}}$
**Output:** Collection of sets $\mathcal{T}$

1: run $\mathcal{A}_{\text{sLD}}$ on $S$ with $\alpha = w_{\text{low}}$ and let $M = \{\mu_1, \ldots, \mu_{|M|}\}$ be the returned list
2: let $v_{ij}$ be a unit vector in the direction of $\mu_i - \mu_j$ for $i \neq j \in [|M|]$
3: $\mathcal{T} \leftarrow \emptyset$ and $R \leftarrow \{1, \ldots, |M|\}$
4: **while** $R \neq \emptyset$ **do**
5:     **for all** $i \in R$ **do**
6:         $S_i^{(1)} \leftarrow \bigcap_{j \in [|M|], j \neq i} \left\{ x \in S, \text{ s.t. } |v_{ij}^\top (x - \mu_i)| \leqslant \gamma + \gamma' \right\}$
7:         $S_i^{(2)} \leftarrow \bigcap_{j \in [|M|], j \neq i} \left\{ x \in S, \text{ s.t. } |v_{ij}^\top (x - \mu_i)| \leqslant 3\gamma + 3\gamma' \right\}$
8:     remove all $i \in R$ for which $|S_i^{(1)}| \leqslant 100 w_{\text{low}}^4 n$
9:     **if** $R = \emptyset$ **then break**
10:     **if** there exists $i \in R$ such that $|S_i^{(2)}| \leqslant 2|S_i^{(1)}|$ **then**
11:         select the $i \in R$ with $|S_i^{(2)}| \leqslant 2|S_i^{(1)}|$ for which $|S_i^{(1)}|$ is largest
12:         $\mathcal{T} \leftarrow \mathcal{T} \cup \left\{ S_i^{(2)} \right\}$
13:         $S \leftarrow S \setminus S_i^{(1)}$
14:         $R \leftarrow R \setminus \{i\}$
15:     **else**
16:         $\mathcal{T} \leftarrow \mathcal{T} \cup \{S\}$
17:         **break**
18: **return** $\mathcal{T}$

---

In words, Theorem B.6 (ii) states that *at initialization*, OuterStage represents each inlier cluster well, i.e., for each $i$, the $i$-th cluster is almost entirely contained in some set $S_j^{(1)}$ for some $j \in [|M|]$. Next, (iii) states that either $S_j^{(1)}$ intersects negligibly some true component, or $S_j^{(2)}$ contains almost entirely the same component. Further, (iv) and (v) state that sets that sufficiently intersect with some true component must be separated from other components and each other.

We now introduce some notation to present the next theorem that establishes further guarantees for the algorithm output during execution. For $\mathcal{T}$, a collection of sets that is the output of Algorithm 6, we define

$$G := \left\{ i \in [k], \text{ such that there exists } T = S_j^{(2)} \in \mathcal{T} \text{ with } \left| S_j^{(1)} \cap C_i^* \right| \geqslant w_{\text{low}}^4 |C_i^*|, \text{ for some } j \right\}. \tag{B.1}$$

In words, it is the set of inlier components for which a corresponding set with "sufficiently many" points from the $i$-th component was added to $\mathcal{T}$. It may happen that for a given index $i \in G$, several $j \in [|M|]$ satisfy $S_j^{(2)} \in \mathcal{T}$ and $\left| S_j^{(1)} \cap C_i^* \right| \geqslant w_{\text{low}}^4 |C_i^*|$. We define $g_i \in [|M|]$ to denote the index of the *first* such set $S_{g_i}^{(2)}$ added to $\mathcal{T}$.

Further, we define $U_i := (C_i^* \cap S_{g_i}^{(2)}) \setminus S_{g_i}^{(1)}$ to be the set of inlier points from the $i$-th component, which were *not* removed from $S$ at the iteration corresponding to $g_i$. Let $U := \cup_{i \in G} U_i$ denote the union of such 'left-over' inlier points.

**Theorem B.7** (Outer stage guarantees, during execution). *Let $S$ consist of $n$ i.i.d. samples from $\mathcal{X}$ as in the statement of Theorem C.1. Run OuterStage (Algorithm 6) on $S$ and consider the moment when the sets $S_i^{(2)}$ are added to $\mathcal{T}$. We have that, with probability at least $1 - w_{\text{low}}^{O(1)}$, all of the following are true:*

  *(i)* $|U| \leqslant (2\varepsilon + O(w_{\text{low}}^2))n$,

  *(ii) for $i \in G$, we have that $\left| S_{g_i}^{(2)} \cap C_i^* \right| \geqslant (1 - \frac{w_{\text{low}}^2}{2} - O(w_{\text{low}}^3)) |C_i^*| \geqslant (1 - w_{\text{low}}^2)w_i n$,*

  *(iii) for $j \in [|M|] \setminus \{g_i \,|\, i \in G\}$, either $\left| S_j^{(2)} \right| \leqslant O(w_{\text{low}}^2)n$, or at least half of the samples in $S_j^{(1)}$ are either adversarial samples or lie in $U$,*

*(iv) if when the else statement is triggered, $|S| \geqslant 0.1w_{\text{low}}n$, then at least a $0.4$-fraction of the samples in $S$ are adversarial, or equivalently, $|S| \leqslant 2.5\varepsilon n$.*

Note that the else statement of $\mathrm{OuterStage}$ can only be triggered once, at the end of the execution. In words, Theorem B.7 (i) states that, for $i \in G$, samples from $i$-th cluster that remained in $S$ after $S_{g_i}^{(1)}$ was removed, constitute a small (comparable with $\varepsilon$) fraction. Further, (ii) states that the sets *added to* $\mathcal{T}$, corresponding to $i \in G$, almost entirely contain $C_i^*$. Finally, (iii) describes the sets that do not correspond to any $g_i, i \in G$. These sets must either be small, or contain a significant amount of outlier points in the neighborhood. The proofs of Theorems B.6 and B.7 can be found in Appendix E.

## C   Proof of Theorem 3.3

In this section, we state and prove a refined version of our main result, Theorem C.1, from which the statement of Theorem 3.3 directly follows.

### C.1   General theorem statement

We define

$$\psi_t(\alpha) = \begin{cases} \sqrt{t}(1/\alpha)^{1/t} & \text{if } t \leqslant 2\log 1/\alpha, \\ \sqrt{2e\log 1/\alpha} & \text{else,} \end{cases} \tag{C.1}$$

which captures a tail decay of a distribution with sub-Gaussian $t$-th central moments: $\mathbb{P}_{x\sim\mathcal{D}}\left(\langle x-\mu, v\rangle^t \geqslant \psi_t(\alpha)\right) \lesssim \alpha$.

We now state our main result for list-decodable mixture learning. Recall that $\varepsilon_{\mathrm{RME}}$ is defined in Assumption 3.1.

**Theorem C.1** (Main mixture model result). *Let $d, k \in \mathbb{N}_+$, $w_{\text{low}} \in (0, 10^{-4}]$, and $t$ be an even integer. For all $i = 1, \ldots, k$, let $D_i(\mu_i)$ be a $d$-dimensional distribution with mean $\mu_i \in \mathbb{R}^d$ and sub-Gaussian $t$-th central moments. Let $\varepsilon > 0$ and, for all $i = 1, \ldots, k$, let $w_i \in [w_{\text{low}}, 1]$, such that $\sum_{i=1}^k w_i + \varepsilon = 1$. Let $\mathfrak{X}$ be the $d$-dimensional mixture distribution*

$$\mathfrak{X} = \sum_{i=1}^k w_i D_i(\mu_i) + \varepsilon Q,$$

*where $Q$ is an unknown adversarial distribution that can depend on all the other parameters. Let $\mathcal{A}_{\mathrm{kLD}}$ and $\mathcal{A}_R$ satisfy Assumption 3.1. Further, suppose that $\|\mu_i - \mu_j\| \geqslant 200\psi_t(w_{\text{low}}^4) + 200f(w_{\text{low}})$ for all $i \neq j \in [k]$.*

*Then there exists an algorithm (Algorithm 2) that, given $\mathrm{poly}(d, 1/w_{\text{low}}) \cdot (N_{LD}(w_{\text{low}}) + N_R(w_{\text{low}}))$ i.i.d. samples from $\mathfrak{X}$, and given also $d$, $k$, $w_{\text{low}}$, and $t$, runs in time $\mathrm{poly}(d, 1/w_{\text{low}}) \cdot (T_{LD}(w_{\text{low}}) + T_R(w_{\text{low}}))$ and with probability at least $1 - w_{\text{low}}^{O(1)}$ outputs a list $L$ of size $|L| \leqslant k + O(\varepsilon/w_{\text{low}})$ such that, for each $i \in [k]$, there exists $\hat{\mu} \in L$ such that:*

$$\|\hat{\mu} - \mu_i\| \leqslant O(\psi_t(\tilde{w}_i/10) + f(\tilde{w}_i/10)), \qquad \text{where} \quad \tilde{w}_i = w_i/(w_i + \varepsilon + w_{\text{low}}^2).$$

*If the relative weight of the $i$-th inlier cluster is large, i.e., $\tilde{w}_i \geqslant 1 - \varepsilon_{\mathrm{RME}} + 2w_{\text{low}}^2$, then there exists $\hat{\mu} \in L$ such that*

$$\|\hat{\mu} - \mu_i\| \leqslant O(g(\tilde{w}_i - 3w_{\text{low}}^2)).$$

Further, we assume $w_{\text{low}} \in (0, 1/10000]$, since this simplifies some of the proofs. We note that in a mixture with $k$ components we necessarily have $w_{\text{low}} \leqslant 1/k$. Furthermore, when $w_{\text{low}} \in (1/10000, 1/2]$, then we obtain the same result by replacing $w_{\text{low}}$ with $w_{\text{low}}/5000$ throughout the statements and the proof. This would only affect both list size and error guarantees by at most a multiplicative constant, which is absorbed in the Big-O notation.

*Proof of Theorem 3.3.* Proof follows directly from Theorem C.1, by noticing that Assumption 3.2 allows to replace $f(\tilde{w}_i/10)$ by $Cf(\tilde{w}_i)$ and $g(\tilde{w}_i - 3w_{\text{low}}^2)$ by $Cg(\tilde{w}_i)$ for some constant $C > 0$ large enough. $\qquad \square$

### C.2 Proof of Theorem C.1

We now show how to use the results on the inner and outer stage, Theorem B.2 and Theorem B.7 respectively, to arrive at the guarantees for the full algorithm Algorithm 2 in Theorem C.1. For simplicity of the exposition, we split the proof of Theorem C.1 into two separate parts, proving that (i) the output list contains an estimate with small error and that (ii) the size of the output list is small. In what follows we condition on the event $E'$ from the proof of Theorem B.7.

**(i) Proof of error statement** We now prove that, conditioned on the event $E$, the list $L$ output by Algorithm 2 for each $i \in [k]$ contains an estimate $\hat{\mu} \in L$, such that,

(1) $\|\hat{\mu} - \mu_i\| \leqslant O(\psi_t(\tilde{w}_i/10) + f(\tilde{w}_i/10))$,

(2) if $\tilde{w}_i \geqslant 1 - \varepsilon_{\mathrm{RME}} + 2w_{\mathrm{low}}^2$, then $\|\hat{\mu} - \mu_i\| \leqslant O(g(\tilde{w}_i - 3w_{\mathrm{low}}^2))$.

We start by showing that list-decoding error guarantees as in (1) are achievable for all inlier clusters and proceed by improving the error to (2) with RME base learner. Recall that $G$ is as defined in Eq. (B.1).

*Proof of (1)* We now show how the output of the base learner and filtering procedure lead to the error in (1). Fix $i \in [k]$. Recall that $C_i$ denotes the set of $w_i n$ points from $i$-th inlier component with mean $\mu_i$.

If $i \in G$, then on event $E$, we have $\left|S_{g_i}^{(2)} \cap C_i^*\right| \geqslant (1 - w_{\mathrm{low}}^2)w_i n$ by Theorem B.7 (ii), $\sum_{j \neq i} \left|S_{g_i}^{(2)} \cap C_j^*\right| \leqslant w_{\mathrm{low}}^4 n$ by Theorem B.6 (iv), and that the total number of adversarial points is at most $(\varepsilon + w_{\mathrm{low}}^4)n$.

Therefore, the fraction of points from $C_i^*$ in $S_{g_i}^{(2)}$ is at least $\frac{(1-w_{\mathrm{low}}^2)w_i}{w_i+\varepsilon+w_{\mathrm{low}}^3}$, which implies $\alpha \geqslant \tilde{w}_i$ as in Definition B.1. Then, by Theorem B.2, the InnerStage algorithm applied to $T$ leads to error $\|\hat{\mu} - \mu_i\| \leqslant O(\psi_t(\tilde{w}_i/4) + f(\tilde{w}_i/4))$. Otherwise, if $i \notin G$, when the OuterStage algorithm reaches the else statement, $S$ contains at least $(1 - O(w_{\mathrm{low}}^3))|C_i^*|$ samples from $C_i^*$. Indeed, since $i \notin G$, each time we remove points from $S$, we remove at most $w_{\mathrm{low}}^4 n$ points from $C_i^*$. By Theorem B.6 (i), we do at most $O(1/w_{\mathrm{low}})$ removals, so when the OuterStage algorithm reaches the else statement, $S$ contains at least $(1 - O(w_{\mathrm{low}}^3))\,|C_i^*|$ samples from $C_i^*$.

We showed that samples from $C_i^*$ make up at least a $(1 - w_{\mathrm{low}}^2)w_i n/|S|$ fraction of $S$. Based on this fact we can then use Theorem B.7 (iv) and the assumption on the range of $w_{\mathrm{low}}$ to conclude that $|S| \leqslant 2.5\varepsilon n$ and that the fraction of inliers is at least $(1 - w_{\mathrm{low}}^2)w_i/(2.5\varepsilon)$. Therefore, $S$ can be seen as containing samples from the corruption model cor-aLD with $\alpha$ at least $w_i/(2.5\varepsilon) \geqslant w_i/(2.5(w_i + \varepsilon))$. Since $S$ is added to $\mathcal{T}$ in the else statement, applying InnerStage yields the error bound as in (1).

*Proof of (2):* Next, we prove that for all inlier components $i$ with large weight, i.e., such that $w_i/(w_i + \varepsilon) \geqslant 1 - \varepsilon_{\mathrm{RME}}$, there exists a set $T \in \mathcal{T}$ that consists of samples from the corruption model cor-aLD with $\alpha \geqslant w_i/(w_i + \varepsilon) - 2w_{\mathrm{low}}^2$. Then, running InnerStage, in particular the RME base learner, results in the error bound as in (2) by Theorem B.2 (ii). If $i \in G$, in the previous paragraph we showed that there exists $T \in \mathcal{T}$, such that the corresponding $\alpha \geqslant \frac{w_i}{w_i+\varepsilon+w_{\mathrm{low}}^3} \geqslant \frac{w_i}{w_i+\varepsilon} - 2w_{\mathrm{low}}^2$. We now prove by contradiction that the case $i \notin G$ does not occur. Now assume $i \notin G$ so that as we argued before when the else statement is triggered, $S$ contains at least $(1 - O(w_{\mathrm{low}}^3))|C_i^*|$ samples from $C_i^*$. By Theorem B.6 (ii), for some $m_i \in [|M|]$, we have that $|S_{m_i}^{(1)} \cap C_i^*| \geqslant (1 - w_{\mathrm{low}}^2/2 - O(w_{\mathrm{low}}^3))|C_i^*|$ and by Theorem B.6 (iv), $S_{m_i}^{(2)}$ contains at most $w_{\mathrm{low}}^4 n$ samples from other true clusters. Then, since $|S_{m_i}^{(2)}| > 2|S_{m_i}^{(1)}|$, we have that $|S_{m_i}^{(2)}|$ contains at least

$$(1 - w_{\mathrm{low}}^2 - O(w_{\mathrm{low}}^3))|C_i^*| - w_{\mathrm{low}}^4 n \geqslant (1 - 1.5w_{\mathrm{low}}^2)|C_i^*|$$

adversarial samples. Therefore, $\varepsilon \geqslant (1 - 1.5w_{\mathrm{low}}^2)|C_i^*|/n$, and using that $|C_i^*| \geqslant w_i n - w_{\mathrm{low}}^{10} n$, we have $\varepsilon \geqslant (1 - 1.5w_{\mathrm{low}}^2)w_i - w_{\mathrm{low}}^{10}$. However, this contradicts $w_i/(w_i + \varepsilon) \geqslant 1 - \varepsilon_{\mathrm{RME}}$ unless $\varepsilon_{\mathrm{RME}} \geqslant 1/2 - 2w_{\mathrm{low}}^2$, which is prohibited by the assumptions in Theorem C.1. Therefore whenever $w_i/(w_i + \varepsilon) \geqslant 1 - \varepsilon_{\mathrm{RME}}$ we are in the case $i \in G$.

**(ii) Proof of small list size** We now prove that on the set $E$, we have that $|L| \leqslant k + O(\varepsilon/w_{\text{low}})$. Here, we need to carefully analyze iterations in the while loop where an inlier component is "selected" for the first time in order to obtain a tight list size bound. Recall that $g_i$ corresponds to the index in $R$ that is first selected for the $i$-th inlier cluster.

*First selection of a component:* For any $i \in [k]$, if $i \in G$, then Theorem B.7 (ii) implies that running InnerStage on $S_{g_i}^{(2)}$ produces a list of size at most $1 + O((|S_{g_i}^{(2)} \setminus C_i^*|)/(w_{\text{low}}n))$. Then, over all $i \in G$, these sets $S_{g_i}^{(2)}$ contribute to the list size $|L|$ at most $k + O\left(\sum_{i=1}^k |S_{g_i}^{(2)} \setminus C_i^*|/(w_{\text{low}}n)\right)$. Furthermore, by Theorem B.6 (v), all these sets $S_{g_i}^{(2)}$ are disjoint and each of them contains at most $w_{\text{low}}^4 n$ samples from other true clusters. Therefore $\sum_{i=1}^k |S_{g_i}^{(2)} \setminus C_i^*| \leqslant \varepsilon n + O(w_{\text{low}}^3)n$. Then the contribution to $|L|$ of all these $S_i$'s corresponding to true clusters is at most $k + O\left((\varepsilon + w_{\text{low}}^3)/w_{\text{low}}\right)$. Note that if $\varepsilon \leqslant w_{\text{low}}^3$ and $w_{\text{low}}$ is small enough, Algorithm 3 actually produces a list of size 1 in each run considered above, so the contribution is exactly $k$; otherwise we can bound the contribution by $k + O(\varepsilon/w_{\text{low}})$.

*Samples left over from a component:* Next, all inlier samples that were not removed, i.e., constituting $U$, can be considered outlier points for the future iterations, which, by Theorem B.7 (i), only increases the outlier fraction to $\tilde{\varepsilon} = 3\varepsilon + O(w_{\text{low}}^2)$. For the same reason as above, without loss of generality, we can consider $\varepsilon > w_{\text{low}}^2$ since otherwise, the corresponding list size overhead (for small enough $w_{\text{low}}^2$) would again amount to zero.

*Clusters of adversarial samples:* For iterations where a set $S_j^{(2)}$ was added to the final list, which does not correspond to some $g_i, i \in G$, Theorem B.7 (iii), states that either (i) at least half of the samples in $S_j^{(2)}$ were adversarial, or (ii) the cardinality of the set on which Algorithm 3 was executed is small. In both cases the set $S_j^{(2)}$ contributes at most $O(\varepsilon/w_{\text{low}})$ to the final list size $|L|$.

*List size in the else statement:* Finally, when the algorithm reaches the else statement, as argued in the first part, by Theorem B.7 (iv), at that iteration $|S| \leqslant O(\varepsilon)n$. Since Algorithm 3 always produces a list of size bounded by $O(|S|/(w_{\text{low}}n))$ (see Remark B.4), the contribution to $|L|$ at this iteration is bounded by $O(\varepsilon/w_{\text{low}})$.

Overall, we obtain the desired bound on $|L|$ of $k + O(\varepsilon/w_{\text{low}})$.

## D Proof of Theorem B.2

**(i) Proof of error statement** We now prove that, with probability $1 - w_{\text{low}}^{O(1)}$, for the output list $L$ of Algorithm 6,

1. there exists $\hat{\mu} \in L$ such that
$$\|\hat{\mu} - \mu^*\| \leqslant O(\psi_t(\alpha/4) + f(\alpha/4)),$$

2. if $\alpha \geqslant 1 - \varepsilon_{\text{RME}}$, then there exists $\hat{\mu} \in L$ such that
$$\|\hat{\mu} - \mu^*\| \leqslant O(g(\alpha - w_{\text{low}}^2)).$$

By Lemma D.1 we have $|M| \leqslant 1/w_{\text{low}}^{O(1)}$ and, with probability at least $1 - w_{\text{low}}^{O(1)}$, there exists $(\hat{\mu}, \hat{\alpha}) \in M$ such that $\hat{\alpha} \geqslant \alpha/4$ and $\|\hat{\mu} - \mu^*\| \leqslant f(\hat{\alpha})$. Then Lemma D.2 implies that, with probability at least $1 - |M|^2 w_{\text{low}}^{O(1)}$, $(\hat{\mu}, \hat{\alpha})$ will not be removed from $M$. Therefore, either $(\hat{\mu}, \hat{\alpha}) \in L$, or there exists $(\tilde{\mu}, \tilde{\alpha}) \in L$ such that (i) $\tilde{\alpha} \geqslant \hat{\alpha}$ and (ii) $\|\tilde{\mu} - \hat{\mu}\| \leqslant 4\beta(\hat{\alpha})$. The latter case implies that $\|\tilde{\mu} - \mu^*\| \leqslant 40\psi_t(\alpha/4) + 4f(\alpha/4)$.

For the second part, set first $\tilde{\mu} = \hat{\mu}$. Then, in the $i^{\text{th}}$ iteration, $\tilde{\mu}$ moves away by at most $(3\tau/2)/2^{i-1}$. Since $\sum_{i=1}^{\infty} 1/2^i \leqslant 1$, the distance between $\tilde{\mu}$ and $\hat{\mu}$ is always bounded by $3\tau$. Now, assume that indeed $\alpha \geqslant 1 - \varepsilon_{\text{RME}}$ and $\|\hat{\mu} - \mu^*\| \leqslant \tau$. Whenever $\tilde{\alpha} \leqslant \alpha$, with high probability $\mathcal{A}_R$ produces some $\mu_{\text{RME}}$ such that $\|\mu_{\text{RME}} - \mu^*\| \leqslant g(\tilde{\alpha}) \leqslant \tilde{\beta}/2$. Furthermore, as long as $\tilde{\alpha} \leqslant \alpha$, at the moment

of the while statement check we have $\|\tilde{\mu} - \mu^*\| \leqslant \tilde{\beta}$: in the first iteration this is by assumption, and in later iterations it follows because $\tilde{\mu}$ is the former $\mu_{\mathrm{RME}}$. Therefore the while statement check passes as long as $\tilde{\alpha} \leqslant \alpha$.

There exists the possibility that the algorithm returns or breaks even though $\tilde{\alpha} \leqslant \alpha$. If the algorithm returns early, then the error $\tau$ achieved by $\hat{\mu}$ is already within a factor of two of the optimal. If the algorithm breaks, either $\tilde{\alpha} + w_{\mathrm{low}}^2 > 1$, case in which $\tilde{\mu}$ already satisfies $\|\tilde{\mu} - \mu^*\| \leqslant g(1 - w_{\mathrm{low}}^2)$, or else $\|\tilde{\mu} - \mu^*\|$ is already within a factor of two of the optimal. Therefore these cases do not affect the error negatively.

Finally, let us consider what happens when $\tilde{\alpha} > \alpha$ and the while statement check continues to pass. The first time we reach some $\tilde{\alpha} > \alpha$, we must have $\|\tilde{\mu} - \mu^*\| \leqslant \tilde{\beta} \leqslant 2g(\alpha - w_{\mathrm{low}}^2)$. Then, in later iterations, $\tilde{\mu}$ can move from this estimate by a distance of most $3\tilde{\beta} \leqslant 6g(\alpha - w_{\mathrm{low}}^2)$, by the same argument as the argument that $\|\tilde{\mu} - \hat{\mu}\| \leqslant 3\tau$. Overall, at the end we have

$$\|\tilde{\mu} - \hat{\mu}\| \leqslant \max(2g(1), g(1 - w_{\mathrm{low}}^2), 8g(\alpha)) \leqslant 8g(\alpha - w_{\mathrm{low}}^2).$$

The number of runs is at most $1/w_{\mathrm{low}}^2$, so with probability $1 - w_{\mathrm{low}}^{O(1)}$ all runs of $\mathcal{A}_R$ succeed.

We showed that there exists $(\hat{\mu}, \hat{\alpha}) \in L$ such that $\hat{\alpha} \geqslant \alpha/4$ and $\|\hat{\mu} - \mu^*\| \leqslant 40\psi_t(\hat{\alpha}) + 4f(\hat{\alpha})$. This error can increase by running ImproveWithRME with $\tau = 40\psi_t(\hat{\alpha}) + 4f(\hat{\alpha})$ to at most

$$\|\hat{\mu} - \mu^*\| \leqslant 160\psi_t(\hat{\alpha}) + 16f(\hat{\alpha}) = O(\psi_t(\alpha) + f(\alpha)).$$

Furthermore, if $\alpha \geqslant 1 - \tau_{\min}$, this $(\hat{\mu}, \hat{\alpha}) \in L$ satisfies the conditions of ImproveWithRME, so with high probability the error is reduced by running ImproveWithRME with $\tau = 40\psi_t(\hat{\alpha}) + 4f(\hat{\alpha})$ to $\|\hat{\mu} - \mu^*\| \leqslant 8g(\alpha - w_{\mathrm{low}}^2)$.

**(ii) Proof of list size**  We now prove that $|L| \leqslant 1 + O((1 - \alpha)/\alpha_{\mathrm{low}})$.

First, assume that $\alpha \leqslant 9/10$. Since all $\hat{\alpha}_s \geqslant \alpha_{\mathrm{low}}$, we have that $|L| \leqslant 10/(9\alpha_{\mathrm{low}}) \leqslant 12(1 - \alpha)/\alpha_{\mathrm{low}}$.

For the rest of the proof we assume that $\alpha > 9/10$. We analyze sets $J$ and $T_i$ for $i \in J$ at the end of execution of ListFilter. In particular, recall that $L = \{(\hat{\mu}_i, \hat{\alpha}_i), \ i \in J\}$. Also, at the end of Algorithm 4 we have the following expression for $T_i$:

$$T_i = \bigcap_{j \in J \setminus \{i\}} \{x \in S, \text{ s.t. } |v_{ij}^\top (x - \hat{\mu}_i)| \leqslant \max(\beta(\hat{\alpha}_i), \beta(\hat{\alpha}_j))\},$$

where $v_{ij}$ are unit vectors in direction $\hat{\mu}_i - \hat{\mu}_j$. Select the $s \in J$ for which $\hat{\alpha}_s$ is maximized. By (i), with probability at least $1 - w_{\mathrm{low}}^{60}$, there exists a hypothesis in $J$ with $\hat{\alpha} \geqslant \alpha/4 \geqslant 0.2$. Then we have that $\hat{\alpha}_s \geqslant 0.2$. In addition, for all hypotheses, $\hat{\alpha}_s \leqslant 1/3$. Let $j \in J$ be such that $j \neq s$. We will show that at least half of the points in $T_j$ are adversarial, i.e., $|T_j| \geqslant 2\,|T_j \cap C^*|$. If this is indeed the case, we can treat all inlier points in all $T_j$ as outliers, as it would at most double total number of outlier points in $S$.

Now, assume that for some $j \neq s$, $|T_j| < 2\,|T_j \cap C^*|$. Note that, because $|T_s| \geqslant 0.9 \cdot 0.2n = 0.18n$ and $|C^*| \geqslant 0.9n$, $|T_s \cap C^*| \geqslant 0.18n - 0.1n \geqslant 0.08n$. Therefore

$$\left|\left\{x \in C^*, \text{ s.t. } |v_{sj}^\top (x - \hat{\mu}_s)| \leqslant \beta(\hat{\alpha}_s)\right\}\right| \geqslant 0.08\,|C^*|.$$

Also note that Lemma H.2 for radius $10\psi_t(1/2) \leqslant 10\psi_t(\hat{\alpha}_s)$ implies that, with exponentially small failure probability,

$$\left|\left\{x \in C^*, \text{ s.t. } |v_{sj}^\top (x - \mu^*)| \leqslant \beta(\hat{\alpha}_s)\right\}\right| \geqslant 0.99\,|C^*|.$$

Since these two sets necessarily intersect, we can bound $|v_{sj}^\top (\hat{\mu}_s - \mu^*)| \leqslant 2\beta(\hat{\alpha}_s)$, implying that $|v_{sj}^\top (\hat{\mu}_j - \mu^*)| \geqslant 2\beta(\hat{\alpha}_j)$, since $\|\hat{\mu}_s - \hat{\mu}_j\| \geqslant 4\beta(\hat{\alpha}_j)$. Thus, if $|v_{sj}^\top (x - \hat{\mu}_j)| \leqslant \beta(\hat{\alpha}_j)$, then $|v_{sj}^\top (x - \mu^*)| > \beta(\hat{\alpha}_j)$, implying that

$$(T_j \cap C^*) \subseteq \left\{x \in C^*, \text{ s.t. } |v_{sj}^\top (x - \mu^*)| > \beta(\hat{\alpha}_j)\right\}. \tag{D.1}$$

However, Lemma H.2 tells us that with high probability only a small fraction of points in $C^*$ satisfies $|v_{sj}^\top (x - \mu^*)| > \beta(\hat{\alpha}_j)$. In particular, applying the lemma with radius $10\psi_t(\hat{\alpha}_j)$, we get that with exponentially small failure probability,

$$\left|\left\{x \in C^*, \text{ s.t. } |v_{sj}^\top (x - \mu^*)| \leqslant \beta(\hat{\alpha}_j)\right\}\right| \geqslant \left(1 - \frac{\hat{\alpha}_j}{50}\right)|C^*|. \tag{D.2}$$

From eqs. (D.1) and (D.2) it follows that $|T_j \cap C^*| \leqslant \hat{\alpha}_j |C^*| / 50$. Using that $|T_j| \geqslant 9\hat{\alpha}_j n/10 \geqslant 9\hat{\alpha}_j |C^*| / 10$ by the properties of ListFilter, we obtain

$$9\hat{\alpha}_j |C^*| / 10 \leqslant |T_j| \leqslant 2 |T_j \cap C^*| \leqslant \hat{\alpha}_j |C^*| / 25,$$

which is a contradiction.

Therefore, for all $j \in J$ such that $j \neq s$, we have that $|T_j| \geqslant 2 |T_j \cap C^*|$. As we said in the beginning, by treating all inlier points in those $T_j$ as outliers we at most double total number of outlier points. Since there are at most $(1 - \alpha + \alpha w_{\text{low}}^2)n$ outlier points and the sets $T_j$ are non-intersecting, we get $\sum_{j \in J \setminus \{s\}} |T_j| \leqslant 2(1 - \alpha + \alpha w_{\text{low}}^2)n$. The lower bound on the size $|T_j| \geqslant 9\alpha_{\text{low}} n/10$ implies $|J/\{s\}| \leqslant \frac{2(1 - \alpha + \alpha w_{\text{low}}^2)n \cdot 10}{9\alpha_{\text{low}} n}$ and thus $|L| = |J| \leqslant 1 + 3(1 - \alpha)/\alpha_{\text{low}}$.

Note that in InnerStage we set $\alpha_{\text{low}} = \min(\alpha_{\text{low}}, 1/100)$. Therefore, for the original $\alpha_{\text{low}}$, the list size is bounded by $1 + 240(1 - \alpha)/\alpha_{\text{low}}$.

**Conclusion** Combining the probabilities of success of all steps, we get that the algorithm succeeds with probability at least $1 - w_{\text{low}}^{O(1)}$ for some large constant in the exponent. Our algorithm, ignoring the calls to $\mathcal{A}_{\text{kLD}}$ and $\mathcal{A}_R$, has sample complexity and time complexity bounded by $\text{poly}(d, 1/w_{\text{low}})$, which gives the desired sample and time complexity when taking $\mathcal{A}_{\text{kLD}}$ and $\mathcal{A}_R$ into consideration. This completes the proof of the theorem.

## D.1 Auxiliary lemmas and proofs

**Lemma D.1** (List initialization). *Let $S$, $\alpha_{\text{low}}$ and $\alpha$ be as in* cor-aLD *model. If* InnerStage *(Algorithm 3) is run with $S$ and $\alpha_{\text{low}}$, the size of $M$ is at most $1/w_{\text{low}}^{O(1)}$, all $(\hat{\mu}, \hat{\alpha}) \in M$ satisfy $\hat{\alpha} \leqslant 1/3$, and with probability at least $1 - w_{\text{low}}^{O(1)}$ there exists $(\hat{\mu}, \hat{\alpha}) \in M$ such that $\alpha/4 \leqslant \hat{\alpha} \leqslant \min(\alpha, 1/3)$ and $\|\hat{\mu} - \mu^*\| \leqslant f(\hat{\alpha})$.*

*Proof.* There are at most $1/\alpha_{\text{low}}$ choices for $\hat{\alpha}$, and for each of them the output of $\mathcal{A}_{\text{kLD}}$ has size at most $1/w_{\text{low}}^{O(1)}$, so $|M| \leqslant 1/w_{\text{low}}^{O(1)}$. With probability $1 - w_{\text{low}}^{O(1)}$, $\mathcal{A}_{\text{kLD}}$ succeeds in all up to $1/\alpha_{\text{low}}$ runs. Then we are guaranteed to produce one $\hat{\alpha}$ with $\alpha/4 \leqslant \hat{\alpha} \leqslant \min(\alpha, 1/3)$, and then $\mathcal{A}_{\text{kLD}}$ is guaranteed to produce one corresponding $\hat{\mu}$ with $\|\hat{\mu} - \mu^*\| \leqslant f(\hat{\alpha})$. $\square$

**Lemma D.2** (Good hypotheses are not removed). *Let $S$, $\alpha_{\text{low}}$ and $\alpha$ be as in* cor-aLD *model. Run* ListFilter *(Algorithm 4) on $S$ and $\alpha_{\text{low}}$ with $M$ obtained from* InnerStage *(Algorithm 3) and call a hypothesis $(\hat{\mu}, \hat{\alpha}) \in M$ good if $\hat{\alpha} \geqslant \alpha/4$ and $\|\hat{\mu} - \mu^*\| \leqslant f(\hat{\alpha})$. Then, with probability at least $|M|^2 w_{\text{low}}^{O(1)}$, no good hypothesis is removed from $M$ (including in any of the reruns triggered by the algorithm).*

*Proof.* Let $\ell$ be an arbitrary iteration of the outer for loop. Then, at the beginning of the $\ell^{\text{th}}$ iteration,

1. $T_j \cap T_s = \emptyset$ for any $j < s \in J$,

2. $|T_j| \geqslant 0.9\hat{\alpha}_j n$ for any $j \in J$,

3. $|J| \leqslant 10/(9\hat{\alpha}_\ell)$.

Indeed, the second property follows directly from the algorithm.

For the first property, assume that for $j < s \in J$, there exists $x \in S$, such that $x \in T_j \cap T_s$. This would imply that $|v_{js}^\top(\hat{\mu}_j - \hat{\mu}_s)| \leqslant 2\beta(\hat{\alpha}_s)$, so $\|\hat{\mu}_j - \hat{\mu}_s\| \leqslant 2\beta(\hat{\alpha}_s)$. However, in this case the first 'if' condition would pass and we would not add $s$ to $J$. Thus, $T_j \cap T_s = \emptyset$.

For the third property, note that

$$n \geqslant \sum_{j \in J} |T_j| \geqslant \sum_{j \in J} 0.9\hat{\alpha}_j n \geqslant 0.9 |J| \hat{\alpha}_\ell n,$$

which implies $|J| \leqslant 10/(9\hat{\alpha}_\ell)$.

Now, let $s$ be the index of a hypothesis with $\hat{\alpha}_s \geqslant \alpha/4$ and $\|\hat{\mu}_s - \mu^*\| \leqslant f(\hat{\alpha}_s)$. If $s$ was skipped in the $s^{\text{th}}$ iteration (i.e., there exists $j \in J$ with $\hat{\mu}_j$ close to $\hat{\mu}_s$), then $(\hat{\mu}_s, \hat{\alpha}_s)$ is trivially not removed from $M$. For the rest of the proof we assume that $s$ is not skipped.

For the sake of the analysis, we introduce the analogue of the sets $T_s$, which we call $\tilde{T}_s$, defined for points in the set $C^*$ (i.e., all inlier points before the adversarial removal), and show that (i) $\left|\tilde{T}_s\right|$ is large and (ii) $\left|\tilde{T}_s \setminus T_s\right|$ is small. In particular, let

$$\tilde{T}_s = \bigcap_{j \in J} \left\{ x \in C^*, \text{ s.t. } |v_{js}^\top (x - \hat{\mu}_s)| \leqslant \beta(\hat{\alpha}_s) \right\},$$

where we recall $\beta(\hat{\alpha}_s) = 10\psi_t(\hat{\alpha}_s) + f(\hat{\alpha}_s)$. Note that $|T_s| \geqslant \left|\tilde{T}_s\right| - |C^* \setminus S^*| \geqslant \left|\tilde{T}_s\right| - w_{\text{low}}^2 |C^*|$. Also, for any $\alpha' \leqslant \hat{\alpha}_s$, applying Lemma H.2 with radius $10\psi_t(\alpha')$, using that $\|\hat{\mu}_s - \mu^*\| \leqslant f(\hat{\alpha}_s) \leqslant f(\alpha')$ and $t \geqslant 2$, we get that with exponentially small failure probability,

$$\left|\left\{ x \in C^*, \text{ s.t. } |v^\top(x - \hat{\mu}_s)| > \beta(\alpha') \right\}\right| \leqslant \frac{\alpha'}{50} |C^*|. \tag{D.3}$$

Consider the $s^{\text{th}}$ iteration. Using a union bound over $|J| \leqslant 2/\alpha_{\text{low}}$ directions, and since all $\hat{\alpha}_s \geqslant \alpha_{\text{low}}$, we get that with exponentially small failure probability

$$\left|\tilde{T}_s\right| \geqslant |C^*| - \sum_{i \in J} \left|\left\{ x \in C^*, \text{ s.t. } |v_{is}^\top(x - \mu_s)| > \beta(\hat{\alpha}_s) \right\}\right| \geqslant \left(1 - \frac{\hat{\alpha}_s}{50}|J|\right)|C^*| \geqslant 0.95\,|C^*|,$$

where we used that and $|J| \leqslant 10/(9\hat{\alpha}_s)$. This implies that

$$|T_s| \geqslant \left|\tilde{T}_s\right| - w_{\text{low}}^2 |C^*| \geqslant (0.95 - w_{\text{low}}^2)|C^*| \geqslant 0.92\,|C^*| \geqslant 0.9\alpha n \geqslant 0.9\hat{\alpha}_s n,$$

i.e., $(\hat{\mu}_s, \hat{\alpha}_s)$ is not removed from $M$ during $s^{\text{th}}$ iteration.

The pair $(\hat{\mu}_s, \hat{\alpha}_s)$ could also be removed during later iterations, when we recalculate $T_s$ by removing points along new directions. However, following a similar argument, we show that still, with high probability, $|T_s| \geqslant 0.9\hat{\alpha}_s n$. Assume that we are now in the $k^{\text{th}}$ iteration of the outer cycle, where $k > s$. We define again $\tilde{T}_s$ and sets $A, B$:

$$\tilde{T}_s := \bigcap_{i \in J \setminus \{s\}} \left\{ x \in C^*, \text{ s.t. } |v_{is}^\top(x - \hat{\mu}_s)| \leqslant \max(\beta(\hat{\alpha}_s), \beta(\hat{\alpha}_i)) \right\},$$

$$A := \bigcap_{i < s, i \in J} A_i, \quad \text{for} \quad A_i := \left\{ x \in C^*, \text{ s.t. } |v_{is}^\top(x - \hat{\mu}_s)| \leqslant \beta(\boldsymbol{\hat{\alpha}_s}) \right\},$$

$$B := \bigcap_{i > s, i \in J} B_i, \quad \text{for} \quad B_i := \left\{ x \in C^*, \text{ s.t. } |v_{is}^\top(x - \hat{\mu}_s)| \leqslant \beta(\boldsymbol{\hat{\alpha}_i}) \right\},$$

so that $\tilde{T}_s = A \cap B$ and again $|T_s| \geqslant \left|\tilde{T}_s\right| - w_{\text{low}}^2 |C^*|$. It is crucial that we have different right hand sides in the definitions of $A_i$ and $B_i$ (we wrote them in boldface to emphasize this).

Using a union bound again, we write

$$\left|\tilde{T}_s\right| \geqslant |C^*| - \sum_{i < s, i \in J} |C^* \setminus A_i| - \sum_{i > s, i \in J} |C^* \setminus B_i|.$$

Using eq. (D.3), with exponentially small failure probability, for all $i \in J$,

$$|C^* \setminus A_i| \leqslant (\hat{\alpha}_s/50)|C^*| \quad \text{(for } i < s) \quad \text{and} \quad |C^* \setminus B_i| \leqslant (\hat{\alpha}_i/50)|C^*| \quad \text{(for } i > s).$$

Next, note that before the last element was added, we had that (i) $T_i \bigcap T_j = \emptyset$ for any $i \neq j \in J$ and (ii) $|T_i| \geqslant 0.9\hat{\alpha}_i n$ for any $i \in J$. This implies that $\sum_{i \in J} \hat{\alpha}_i < 10/9 + \hat{\alpha}_{\text{last}} < 19/9$, where $\hat{\alpha}_{\text{last}}$ corresponds to the element which was added last (it might happen that after addition of the last element, we have $|T_i| < 0.9\hat{\alpha}_i n$ for several $i \in J$). Therefore, as before, we obtain that

$$\left|\tilde{T}_s\right| \geqslant \left(1 - \sum_{i < s, i \in J} (\hat{\alpha}_s/50) - \sum_{i > s, i \in J} (\hat{\alpha}_i/50)\right)|C^*| \geqslant (1 - 10/(9 \cdot 50) - 19/(9 \cdot 50))|C^*| \geqslant 0.93\,|C^*|,$$

therefore $|T_s| \geqslant \left|\tilde{T}_s\right| - w_{\text{low}}^2 |C^*| \geqslant 0.92 |C^*| \geqslant 0.9\hat{\alpha}_s n$ and $(\hat{\mu}_s, \hat{\alpha}_s)$ will not be removed from $M$.

We established that in a single run of the algorithm a good hypothesis is removed with exponentially small probability. The number of good hypotheses is bounded by $|M|$. Furthermore, the number of runs of the algorithm is also bounded by $|M|$, since whenever the algorithm is rerun a hypothesis is removed from $M$. Then, by a union bound, we can bound the probability that any good hypothesis is removed in any run of the algorithm by $|M|^2 \, w_{\text{low}}^{O(1)}$. $\qquad\square$

# E   Proof of outer stage algorithm guarantees in Appendix B.3

Recall that $\gamma = 4\psi_t(w_{\text{low}}^4)$ and $\gamma' = 160\psi_t(w_{\text{low}}/4) + 16f(w_{\text{low}}/4)$.

## E.1   Proof of Theorem B.6

In what follows we condition on the event $E$ that the events under which the conclusions in Lemmas H.2 and H.3 hold and that $\mathcal{A}_{\text{sLD}}$ succeeds. This event holds with probability $1 - w_{\text{low}}^{O(1)}$ by Assumption 3.1, Remark B.3 and union bound (also see Appendix G).

**Proof of Theorem B.6 (i)**   The list size bound follows from the standard results on $\mathcal{A}_{\text{sLD}}$ (see [3], Proposition B.1).

**Proof of Theorem B.6 (ii)**   Guarantees of $\mathcal{A}_{\text{sLD}}$ imply that there exists $\mu_i \in M$ such that $\|\mu_i - \mu^*\| \leqslant \gamma'$. By Lemma H.3, a $(1 - w_{\text{low}}^2/2)$-fraction of the samples in $C^*$ are $\gamma$-close to $\mu^*$ along each direction $v_{ij}$ with $i \neq j \in [|M|]$. Then, the same $(1 - w_{\text{low}}^2/2)$-fraction of samples are $(\gamma + \gamma')$-close to $\mu_i$ along each direction $v_{ij}$, so they are included in $S_i^{(1)}$.

**Proof of Theorem B.6 (iii)**   Suppose $|S_i^{(1)} \cap C^*| \geqslant w_{\text{low}}^4 |C^*|$. Previous point implies that there exists $\mu_j \in M$ be such that $\|\mu_j - \mu^*\| \leqslant \gamma'$. Then at least an $w_{\text{low}}^4$-fraction of the samples in $C^*$ are $(\gamma + \gamma')$-close to $\mu_i$ in direction $\mu_i - \mu_j$. By Lemma H.2, $\mu^*$ is also $\gamma$-close in direction $\mu_i - \mu_j$ to more than a $(1 - w_{\text{low}}^4)$-fraction of the samples in $C^*$, so it is $\gamma$-close to at least one sample in any $w_{\text{low}}^4$-fraction of samples in $C^*$. Therefore $\mu^*$ is also $(2\gamma + \gamma')$-close to $\mu_i$ in direction $\mu_i - \mu_j$. Then $\|\mu_i - \mu_j\| \leqslant 2\gamma + 2\gamma'$ and $\|\mu_i - \mu^*\| \leqslant 2\gamma + 3\gamma'$. Again, using Lemma H.3 we obtain that there exists a $(1 - w_{\text{low}}^2/2)$-fraction of the samples in $C^*$, which is included in $S_i^{(2)}$.

**Proof of Theorem B.6 (iv)**   Similarly, if $|S_i^{(2)} \cap C^*| \geqslant w_{\text{low}}^4 |C^*|$, then there exists $\mu_j \in M$, such that at least an $w_{\text{low}}^4$-fraction of the samples in $C^*$ are $(3\gamma + 3\gamma')$-close to $\mu_i$ in direction $\mu_i - \mu_j$. By the same arguments as in previous paragraph, we obtain that $\|\mu_i - \mu_j\| \leqslant 4\gamma + 4\gamma'$ and $\|\mu_i - \mu^*\| \leqslant 4\gamma + 5\gamma'$.

Then any other true cluster with mean $(\mu^*)'$ and set of samples $(C^*)'$ satisfies $\|\mu^* - (\mu^*)'\| \geqslant 16\gamma + 16\gamma'$, so $\|\mu_i - (\mu^*)'\| \geqslant 12\gamma + 11\gamma'$. From guarantees of $\mathcal{A}_{\text{sLD}}$, there exists $\mu_j' \in M$ such that $\|\mu_j' - (\mu^*)'\| \leqslant \gamma'$. Then $\|\mu_i - \mu_j'\| \geqslant 12\gamma + 10\gamma'$. By Lemma H.2, more than an $w_{\text{low}}^4$-fraction of the samples from $(C^*)'$ are $\gamma$-close to $(\mu^*)'$ in direction $\mu_i - \mu_j'$, so also $(\gamma + \gamma')$-close to $\mu_j'$ in direction $\mu_i - \mu_j'$, so also $(11\gamma + 9\gamma')$-far from $\mu_i$ in direction $\mu_i - \mu_j'$. Then $S_i^{(2)}$ selects at most a $w_{\text{low}}^4$-fraction of the samples from $(C^*)'$. Overall, $S_i^{(2)}$ selects from all other true clusters at most $w_{\text{low}}^4 n$ samples.

**Proof of Theorem B.6 (v)**   Note that by the same argument, $\|\mu_i - \mu^*\| \leqslant 4\gamma + 5\gamma'$ and $\|\mu_{i'} - (\mu^*)'\| \leqslant 4\gamma + 5\gamma'$. However, $\|\mu^* - (\mu^*)'\| \geqslant 16\gamma + 16\gamma'$, so also $\|\mu_i - \mu_{i'}\| \geqslant 8\gamma + 6\gamma'$, so $S_i^{(2)}$ and $S_{i'}^{(2)}$ are disjoint by the condition that each selects only samples that are $(3\gamma + 3\gamma')$-close along direction $\mu_i - \mu_{i'}$ to the respective means $\mu_i$ and $\mu_{i'}$.

## E.2  Proof of Theorem B.7

In the sequel, for any $i \in G$, let $m_i$ be the index in $R$ after initialization that satisfies Theorem B.6 (ii). We condition on the event $E'$ that event $E$ from the proof of Theorem B.6 holds and that both $||C_i^*| - w_i n| \leqslant w_{\text{low}}^{10} n$ for all $i \in [k]$ and the number of adversarial points lies in the range $\varepsilon n \pm w_{\text{low}}^{10} n$. By Hoeffding's inequality and the union bound, the probability of $E'$ is at least $1 - w_{\text{low}}^{O(1)}$.

**Proof of Theorem B.7 (i)**  Let $i \in G$, and consider the beginning of the iteration when $\mu_{g_i}$ is selected. Then, using that all previous iterations could have removed at most $O(w_{\text{low}}^3)|C_i^*|$ samples from $C_i^*$, we have that

$$|S_{m_i}^{(1)} \cap C_i^*| \geqslant (1 - w_{\text{low}}^2/2 - O(w_{\text{low}}^3))|C_i^*|.$$

Therefore at the iteration in which $\mu_{g_i}$ is selected, we still have $m_i \in R$. We now discuss two cases: First, consider the case that $|S_{m_i}^{(2)}| \leqslant 2|S_{m_i}^{(1)}|$. Then, because we selected $\mu_{g_i} \in M$ and not $\mu_{m_i} \in M$ it means that $|S_{g_i}^{(1)}| \geqslant |S_{m_i}^{(1)}| \geqslant (1 - w_{\text{low}}^2/2 - O(w_{\text{low}}^3))|C_i^*|$. Note also by Theorem B.6 (iv), the number of samples from other true clusters in $S_{g_i}^{(2)}$ is at most $w_{\text{low}}^4 n$. Then the number of adversarial samples in $S_{g_i}^{(2)}$ is at least

$$|S_{g_i}^{(2)}| - |C_i^*| - w_{\text{low}}^4 n \geqslant |S_{g_i}^{(2)} \setminus S_{g_i}^{(1)}| - O(w_{\text{low}}^2)|C_i^*| - w_{\text{low}}^4 n \geqslant |S_{g_i}^{(2)} \setminus S_{g_i}^{(1)}| - O(w_{\text{low}}^2)|C_i^*|.$$

Then, either $\left|S_{g_i}^{(2)} \setminus S_{g_i}^{(1)}\right| = O(w_{\text{low}}^2)|C_i^*|$ and $|U_i| \leqslant \left|S_{g_i}^{(2)} \setminus S_{g_i}^{(1)}\right| = O(w_{\text{low}}^2)|C_i^*|$, or $\left|S_{g_i}^{(2)} \setminus S_{g_i}^{(1)}\right| \gg w_{\text{low}}^2|C_i^*|$. In the latter case, even if $S_{g_i}^{(2)} \setminus S_{g_i}^{(1)}$ consists of adversarial examples only, then, since $|S_{g_i}^{(2)}| \leqslant 2|S_{g_i}^{(1)}|$, $U_i$ contains at most double the number of adversarial examples in $S_{g_i}^{(1)}$, i.e. $|U_i| \leqslant 2V_i$ where $V_i$ denotes the number of adversarial examples in $S_{g_i}^{(1)}$.

Now consider the case that $|S_{m_i}^{(2)}| > 2|S_{m_i}^{(1)}|$. By Theorem B.6 (iv), the number of samples from true clusters in $S_{m_i}^{(2)}$ is at most $|C_i^*| + w_{\text{low}}^4 n \leqslant 1.02|S_{m_i}^{(1)}|$, so the number $W_i$ of adversarial samples in $S_{m_i}^{(2)}$ is at least $W_i \geqslant |S_{m_i}^{(2)}| - 1.02|S_{m_i}^{(1)}| \geqslant 0.98|S_{m_i}^{(1)}| \geqslant 0.96|C_i^*|$. Then, $|U_i| = \left|(C_i^* \cap S_{g_i}^{(2)}) \setminus S_{g_i}^{(1)}\right| \leqslant |C_i^*| \leqslant 2W_i$.

Finally note that by Theorem B.6 (v), the sets $S_{g_i}^{(2)}$ and $S_{m_i}^{(2)}$ are disjoint from any other sets $S_{g_j}^{(2)}$ and $S_{m_j}^{(2)}$ that correspond to another component $C_j^*$. Therefore, the number of adversarial examples in the $S_{m_i}^{(2)}$ in the second case and $S_{g_i}^{(2)}$ in the first case is smaller than the total number of adversarial examples, i.e.

$$\sum_{\substack{i \in G \\ |S_{m_i}^{(2)}| \leqslant 2|S_{m_i}^{(1)}|}} V_i + \sum_{\substack{i \in G \\ |S_{m_i}^{(2)}| > 2|S_{m_i}^{(1)}|}} W_i \leqslant (\varepsilon + w_{\text{low}}^{10})n.$$

Therefore, we directly obtain

$$|U| \leqslant \sum_{i \in G} \left|(C_i^* \cap S_{g_i}^{(2)}) \setminus S_{g_i}^{(1)}\right| = \sum_{\substack{i \in G \\ |S_{m_i}^{(2)}| \leqslant 2|S_{m_i}^{(1)}|}} \left|(C_i^* \cap S_{g_i}^{(2)}) \setminus S_{g_i}^{(1)}\right| + \sum_{\substack{i \in G \\ |S_{m_i}^{(2)}| > 2|S_{m_i}^{(1)}|}} \left|(C_i^* \cap S_{g_i}^{(2)}) \setminus S_{g_i}^{(1)}\right|$$

$$\leqslant \sum_{\substack{i \in G \\ |S_{m_i}^{(2)}| \leqslant 2|S_{m_i}^{(1)}|}} 2V_i + \sum_{\substack{i \in G \\ |S_{m_i}^{(2)}| > 2|S_{m_i}^{(1)}|}} 2W_i + O(w_{\text{low}}^2)n \leqslant (2\varepsilon + O(w_{\text{low}}^2))n.$$

**Proof of Theorem B.7 (ii)**  Each iteration before $g_i$ was selected, removed at most $w_{\text{low}}^4|C_i^*|$ samples from $C_i^*$, so all previous iterations removed at most $O(w_{\text{low}}^3)|C_i^*|$ samples from $C_i^*$. Then, by Lemma B.6 (iii), $S_{g_i}^{(2)}$ contains at least $(1 - w_{\text{low}}^2/2 - O(w_{\text{low}}^3))|C_i^*|$ samples from $C_i^*$. The statement follows then since on the event $E'$, we have $w^* n - w_{\text{low}}^{10} n \leqslant |C^*| \leqslant w^* n + w_{\text{low}}^{10} n$.

**Proof of Theorem B.7 (iii)** Here, either for all $i \in [k]$, $|S_j^{(1)} \cap C_i^*| < w_{\text{low}}^4 |C_i^*|$ or $i \in G$ and the algorithm had already selected in a previous iteration $\mu_{g_i} \in M$ with $|S_{g_i}^{(1)} \cap C_i^*| \geqslant w_{\text{low}}^4 |C_i^*|$. Consider a first case, in which $|S_j^{(1)} \cap C_i^*| < w_{\text{low}}^4 |C_i^*|$ for all $i \in [k]$. Then the total number of samples from true clusters in $S_j^{(1)}$ is at most $w_{\text{low}}^4 n$. Using that $|S_j^{(1)}| > 100 w_{\text{low}}^4 n$, it follows that more than half of the samples in $S_j^{(1)}$ are adversarial.

The second case is that $|S_j^{(1)} \cap C_i^*| \geqslant w_{\text{low}}^4 |C_i^*|$ for some $i \in G$ for which in a previous iteration $g_i$ we had that $|S_{g_i}^{(1)} \cap C_i^*| \geqslant w_{\text{low}}^4 |C_i^*|$. Note that at most $w_{\text{low}}^2 |C_i^*|/2$ of the samples in $S \cap C_i^*$ are not considered adversarial at this point (the ones that were outside $S_{g_i}^{(2)}$). Also, by Theorem B.6 (iv), $S_j^{(1)}$ contains at most $w_{\text{low}}^4 n$ samples from other true clusters. Therefore either more than half of the samples in $S_j^{(1)}$ are considered adversarial or

$$|S_j^{(1)}| \leqslant w_{\text{low}}^2 |C_i^*| + 2 w_{\text{low}}^4 n \leqslant O(w_{\text{low}}^2) n.$$

**Proof of Theorem B.7 (iv)** Suppose that when the algorithm reaches the else statement we have for some $i \in [k]$ that $i \in R$ and $|S_{m_i}^{(1)} \cap C_i^*| \geqslant 20 w_{\text{low}}^2 |C_i^*|$. We have that $|S_{m_i}^{(2)} \cap C_i^*|$ is at most $|S_{m_i}^{(1)} \cap C_i^*| + w_{\text{low}}^2 |C_i^*|/2$, where we use that by Theorem B.6 (ii), at most $w_{\text{low}}^2 |C_i^*|/2$ samples can fail to be captured by $S_{m_i}^{(1)}$. By Theorem B.6 (iv), furthermore, the number of samples from other true clusters in $S_{m_i}^{(2)}$ is at most $w_{\text{low}}^4 n$. Therefore, using that $|S_{m_i}^{(2)}| > 2|S_{m_i}^{(1)}|$, the number of adversarial samples in $S_{m_i}^{(2)}$ is at least

$$|S_{m_i}^{(2)}| - |S_{m_i}^{(1)} \cap C_i^*| - w_{\text{low}}^2 |C_i^*|/2 - w_{\text{low}}^4 n \geqslant 0.45 |S_{m_i}^{(2)}| - w_{\text{low}}^4 n \geqslant 0.44 |S_{m_i}^{(2)}|,$$

where in the last inequality we used that $|S_{m_i}^{(2)}| > 100 w_{\text{low}}^4 n$. Let $V$ be the union, over all $i \in [k]$, of all sets $S_{m_i}^{(2)}$ such that $i \in R$ and $|S_{m_i}^{(1)} \cap C_i^*| \geqslant 20 w_{\text{low}}^2 |C_i^*|$. Theorem B.6 (v) gives that all such sets $S_{m_i}^{(2)}$ are disjoint. Therefore at least a $0.44$-fraction of the samples in $V$ are adversarial.

Consider now for some $i \in [k]$ how many samples from $S \cap C_i^*$ can be outside $V$ when the algorithm reaches the else statement. By Theorem B.6 (ii), $S_{m_i}^{(1)}$ can fail to capture at most $w_{\text{low}}^2 |C_i^*|/2$ samples from $C_i^*$, and we have no guarantee that these samples are in $V$. Consider now the samples in $S_{m_i}^{(1)} \cap C_i^*$. If $i \in R$, we may miss up to $20 w_{\text{low}}^2 |C_i^*|$ of these samples if $|S_{m_i}^{(1)} \cap C_i^*| < 20 w_{\text{low}}^2 |C_i^*|$, because in this case we do not include $S_{m_i}^{(2)}$ in $V$. On the other hand, if $i \notin R$, there are at most $100 w_{\text{low}}^4 n$ samples in $S_{m_i}^{(1)} \cap C_i^*$. Then the total number of samples from $S \cap C_i^*$ outside $V$ is at most $w_{\text{low}}^2 |C_i^*|/2 + 20 w_{\text{low}}^2 |C_i^*| + 100 w_{\text{low}}^4 n$. Summed across all $i \in [k]$, this makes up at most $21 w_{\text{low}}^2 n$ samples.

Overall, the number of adversarial samples in $S$ when the algorithm reaches the else statement is at least

$$0.44|V| + (|S| - |V| - 21 w_{\text{low}}^2 n) = |S| - 0.56|V| - 21 w_{\text{low}}^2 n \geqslant 0.44|S| - 21 w_{\text{low}}^2 n \geqslant 0.4|S|$$

where in the last inequality we also used that $|S| \geqslant 0.1 w_{\text{low}} n$.

# F   Proof of  Proposition 3.5

We now prove lower bounds for the case of Gaussian distributions and distributions with $t$-th sub-Gaussian moments.

## F.1   Case b): For the Gaussian inliers

We first focus on the case when $D_i(\mu_i) = \mathcal{N}(\mu_i, I)$. The proof goes through an efficient reduction from the problem considered by Proposition F.1 to the problem solved by algorithm $\mathcal{A}$.

**Proposition F.1** ([5], Proposition 5.11). *Let $\mathcal{D}$ be the class of identity covariance Gaussians on $\mathbb{R}^d$ and let $0 < \alpha \leqslant 1/2$. Then any list-decoding algorithm that learns the mean of an element of $\mathcal{D}$ with failure probability at most $1/2$, given access to $(1 - \alpha)$-additively corrupted samples, must either have error bound $\beta = \Omega(\sqrt{\log 1/\alpha})$ or return $\min(2^{\Omega(d)}, (1/\alpha)^{w(1)})$ many hypotheses.*

First, we describe the means of the components in the input distribution to algorithm $\mathcal{A}$. Let $\bar{\mu}_1, \ldots, \bar{\mu}_{k-1} \in \mathbb{R}^d$ be any set of $k - 1$ points with pairwise separation larger than $2C\sqrt{\log 1/w_{\text{low}}}$. Then let $\mu_k = (\bar{\mu}, 0) \in \mathbb{R}^{d+1}$ and $\mu_i = (\bar{\mu}_i, 2C\sqrt{\log 1/w_{\text{low}}} + 1) \in \mathbb{R}^{d+1}$ for all $i \in [k-1]$. Then $\mu_1, \ldots, \mu_k$ also have pairwise separation larger than $2C\sqrt{\log 1/w_{\text{low}}}$.

Then, given $n$ points $y_1, \ldots, y_n \in \mathbb{R}^d$ as in the input to the problem in Proposition F.1 (i.e. $(1 - \alpha)$-additively corrupted samples), we generate $n$ points that we give as input to algorithm $\mathcal{A}$ as follows: let $S = \{1, \ldots, n\}$, and then for each of the $n$ points, draw $i \sim \text{Unif}\{1, \ldots, k\}$ and generate the point as follows:

1. if $i \in [k-1]$, sample the point from $N(\mu_i, I_{d+1})$,

2. if $i = k$, sample $j \sim S$ uniformly at random, remove $j$ from $S$, sample $g \sim N(0, 1)$, and let the point be $(y_j, g) \in \mathbb{R}^{d+1}$.

We note that this construction simulates an input sampled i.i.d. according to the mixture $\frac{1}{k}N(\mu_1, I_{d+1}) + \ldots + \frac{1}{k}N(\mu_{k-1}, I_{d+1}) + \frac{\alpha}{k}N(\mu_k, I_{d+1}) + \frac{1-\alpha}{k}Q'$ for some $Q'$. Then with success probability at least $1/2$ running $\mathcal{A}$ on this input with $w_{\text{low}} = \frac{\alpha}{k}$ returns a list $L$ such that there exists $\hat{\mu} \in L$ with $\|\hat{\mu} - \mu_k\| \leqslant \beta_k$. Note that this implies that $\|(\hat{\mu})_{1:d} - \bar{\mu}\| \leqslant \beta_k$. Finally, we create a pruned list $L'$ as follows: initialize $L' = L$ and then for each $i \in [k-1]$ remove all $\hat{\mu} \in L'$ such that $\|\hat{\mu} - \mu_i\| \leqslant C\sqrt{\log 1/w_{\text{low}}}$. Then we return $L'$ as the output for the original problem in Proposition F.1.

Let us analyze now this output. The separation between the means ensures that any hypothesis $\hat{\mu} \in L$ that is $C\sqrt{\log 1/w_{\text{low}}}$-close to $\mu_k$ is not removed in the pruning. Therefore $L'$ continues to contain a hypothesis $\hat{\mu}$ such that $\|(\hat{\mu})_{1:d} - \bar{\mu}\| \leqslant \beta_k$. Then, if $\beta_k \neq \Omega(\sqrt{\log 1/\alpha})$ and $|L'| < \min\{2^{\Omega(d)}, ((w_k + \varepsilon)/w_k)^{\omega(1)}\}$, this reduction violates the lower bound of Proposition F.1. Therefore we must have either $\beta_k = \Omega(\sqrt{\log 1/\alpha})$ or $|L'| \geqslant \min\{2^{\Omega(d)}, (1/\tilde{w}_k)^{\omega(1)}\}$.

Finally, we show that these lower bounds on $\beta_k$ and $|L'|$ imply the desired lower bound for $\mathcal{A}$. Consider first the case: $\beta_k = \Omega(\sqrt{\log 1/\alpha})$. Note that in the input to algorithm $\mathcal{A}$ we have $\tilde{w}_k = \alpha$. Therefore $\beta_k = \Omega(\sqrt{\log 1/\alpha})$ corresponds to the desired lower bound in the lemma statement. Consider second the case: $|L'| \geqslant \min\{2^{\Omega(d)}, (1/\tilde{w}_k)^{\omega(1)}\}$. We note that, for each $i \in [k-1]$, the original list $L$ must contain some $\hat{\mu} \in L$ such that $\|\hat{\mu} - \mu_i\| \leqslant C\sqrt{\log 1/w_{\text{low}}}$. Furthermore, because the means $\mu_i$ have pairwise separation larger than $2C\sqrt{\log 1/w_{\text{low}}}$, the original list $L$ must contain at least $k - 1$ means of this kind. However, all of these means are removed in the pruning procedure, so $|L| \geqslant k - 1 + |L'|$, so $|L| \geqslant k - 1 + \min\{2^{\Omega(d)}, (1/\tilde{w}_k)^{\omega(1)}\}$. This matches the desired lower bound in the lemma statement. (The choice to make the hidden mean the $k$-th mean was without loss of generality, as the distribution is invariant to permutations of the components.)

## F.2 Case a): For distributions with $t$-th sub-Gaussian moments

The proof for the case when $D_i(\mu_i)$ has sub-Gaussian $t$-th central moments employs the same reduction scheme, but reduces from Proposition F.2.

**Proposition F.2** ([5], Proposition 5.12). *Let $\mathcal{D}$ be the class of distributions on $\mathbb{R}^d$ with bounded $t$-th central moments for some positive even integer $t$, and let $0 < \alpha < 2^{-t-1}$. Then any list-decoding algorithm that learns the mean of an element of $\mathcal{D}$ with failure probability at most $1/2$, given access to $(1 - \alpha)$-additively corrupted samples, must either have error bound $\beta = \Omega(\alpha^{-1/t})$ or return a list of at least $d$ hypotheses.*

Furthermore, in [3], formal evidence of computational hardness was obtained (see their Theorem 5.7, which gives a lower bound in the statistical query model introduced by [14]) that suggests obtaining error $\Omega_t((1/\tilde{w}_s)^{1/t})$ requires running time at least $d^{\Omega(t)}$. This was proved for Gaussian inliers and the running time matches ours up to a constant in the exponent.

# G  Stability of list-decoding algorithms

In this section we discuss two of the existing list-decodable mean estimation algorithms for identity-covariance Gaussian distributions and show that they also work when a $w_{\mathrm{low}}^2$-fraction of the inliers is adversarially removed.

First, we consider the algorithm in Theorem 3.1 in [3]. A central object in their analysis is an "$\alpha$-good multiset", which is a multiset of samples such that all are within distance $O(\sqrt{d})$ of each other and at least an $\alpha$-fraction of them come from a $(1 - \Omega(\alpha))$-fraction of an i.i.d. set of samples from a Gaussian distribution $N(\mu, I_d)$. Then their algorithm essentially works as long as the input contains an $\alpha$-good multiset. For our case, after the removal of a $w_{\mathrm{low}}^2$-fraction of inliers, the input essentially continues to contain a $(1 - w_{\mathrm{low}}^2)\alpha$-good multiset, so the algorithm continues to work in our corruption model.

Second, we consider the algorithm in Theorem 6.12 in [5]. The main distributional requirement of their algorithm is that $\mathbb{E}_{x,y \sim S^*}[p^2(x - y)] \leqslant 2\mathbb{E}_{g,h \sim N(0,I_d)}[p^2(g - h)]$ for all degree-$(t/2)$ polynomials $p$, where $S^*$ is the set of inliers. Concentration arguments give with high probability that $\mathbb{E}_{x,y \sim C^*}[p^2(x - y)] \leqslant 1.5\mathbb{E}_{g,h \sim N(0,I_d)}[p^2(g - h)]$. Furthermore, the distribution over $x, y \sim S^*$ can be seen as a $(1 - w_{\mathrm{low}}^2)^2$-fraction of the distribution over $x, y \sim C^*$. Then Fact G.1, which follows by standard probability calculations, also gives that any event under the former distribution can be bounded in terms of the second distribution:

**Fact G.1.** For any event $A$,

$$\mathop{\mathbb{P}}_{x,y \sim S^*}(A) \leqslant \mathop{\mathbb{P}}_{x,y \sim C^*}(A)/(1 - w_{\mathrm{low}}^2)^2, \tag{G.1}$$

where probabilities are taken over a uniform sample from $S^*$ and $C^*$ respectively.

Overall we obtain

$$\mathbb{E}_{x,y \sim S^*}[p^2(x - y)] \leqslant 1.5/(1 - w_{\mathrm{low}}^2)^2 \mathbb{E}_{g,h \sim N(0,I_d)}[p^2(g - h)],$$

so for $w_{\mathrm{low}}$ small enough we have $\mathbb{E}_{x,y \sim S^*}[p^2(x-y)] \leqslant 2\mathbb{E}_{g,h \sim N(0,I_d)}[p^2(g-h)]$ and their algorithm continues to work in our corruption model.

# H  Concentration bounds

In this section we prove some concentration bounds essential to our analysis.

**Lemma H.1.** *Let $D$ be a $d$-dimensional distribution with mean $\mu^* \in \mathbb{R}^d$ and sub-Gaussian $t$-th central moments with parameter 1. Fix a unit vector $v \in \mathbb{R}^d$. Then*

$$\mathop{\mathbb{P}}_{x \sim D}[|\langle x - \mu^*, v \rangle| \leqslant R] \geqslant 1 - \left(\frac{\sqrt{t}}{R}\right)^t.$$

*Proof.* We have that

$$\mathop{\mathbb{P}}_{x \sim D}[|\langle x - \mu^*, v \rangle| > R] \leqslant \frac{\mathbb{E}_{x \sim D}\langle x - \mu^*, v \rangle^t}{R^t} \leqslant \frac{(t-1)!!}{R^t} \leqslant \left(\frac{\sqrt{t}}{R}\right)^t,$$

where we used that $(t - 1)!! \leqslant t^{t/2} = \sqrt{t}^t$. $\square$

**Lemma H.2.** *Let $D$ be a $d$-dimensional distribution with mean $\mu^* \in \mathbb{R}^d$ and sub-Gaussian $t$-th central moments with parameter 1. Let $C^*$ be a set of i.i.d. samples drawn from $D$. Fix a unit vector $v \in \mathbb{R}^d$. Then with probability at least $1 - \exp\left(-2|C^*| \left(\frac{\sqrt{t}}{R}\right)^{2t}\right)$,*

$$|\{x \in C^*, \text{ s.t. } |\langle x - \mu^*, v \rangle| \leqslant R\}| \geqslant \left(1 - 2\left(\frac{\sqrt{t}}{R}\right)^t\right)|C^*|.$$

*Proof.* The result follows by Lemma H.1 and a Binomial tail bound. $\square$

**Lemma H.3.** *Let $D$ be a $d$-dimensional distribution with mean $\mu^* \in \mathbb{R}^d$ and sub-Gaussian $t$-th central moments with parameter $1$. Let $C^*$ be a set of i.i.d. samples drawn from $D$. Fix $m$ unit vectors $v_1, \ldots, v_m \in \mathbb{R}^d$. Then with probability at least $1 - \exp\left(-2|S^*|m^2\left(\frac{\sqrt{t}}{R}\right)^{2t}\right)$,*

$$\left| \bigcap_{i \in [m]} \{x \in S^*, \text{ s.t. } |\langle x - \mu^*, v_i\rangle| \leqslant R\} \right| \geqslant \left(1 - 2m\left(\frac{\sqrt{t}}{R}\right)^t\right)|S^*|.$$

*Proof.* By Lemma H.1 and a union bound over the $m$ directions, we get

$$\mathbb{P}_{x \sim D}[|\langle x - \mu^*, v_i\rangle| \leqslant R, \forall i \in [m]] \geqslant 1 - m\left(\frac{\sqrt{t}}{R}\right)^t.$$

Then the result follows by a Binomial tail bound. $\qquad\square$

## I  Experimental details

**Adversarial line and adversarial clusters**  The following figure illustrates the adversarial distributions used in Figure 2 and further in this section.

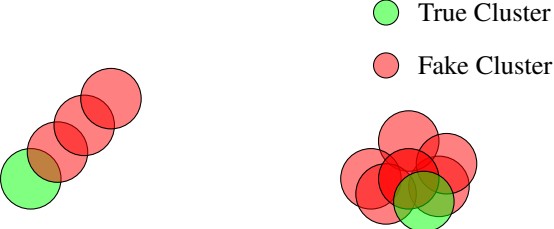

Figure 4: Two variants of adversarial distribution: adversarial line (left) and adversarial clusters (right).

**Data Distribution**  We consider a mixture of $k = 7$ well-separated ($\|\mu_i - \mu_j\| \geqslant 40$) $d = 100$ dimensional inlier clusters whose subgroup sizes range from $0.3$ to $0.02$. The experiments are conducted once using a Gaussian distribution and once using a heavy-tailed t-distribution with five degrees of freedom for both inlier and adversarial clusters. In Figure 6 the latter suggests that our algorithm works comparatively well even for mixture distributions which do not fulfill our assumptions. We set $w_{\text{low}} = 0.02$ and $\varepsilon = 0.12$ so that it is larger than the smallest clusters but smaller than the largest ones and set the total number of data points to $10000$. The Gaussian noise model simply computes the empirical mean and covariance matrix of the clean data and samples $1200$ noisy samples from a Gaussian distribution with this mean and covariance. The adversarial cluster model and the adversarial model are as depicted in Figure 4.

**Attack distributions**  We consider three distinct adversarial models (see Figure 4 for reference).

1. *Adversarial clusters*: After sampling the inlier cluster means, we choose the cluster with the smallest weight. Let $\mu_s$ denote its mean. Then, we sample a random direction $v_c$ with $\|v_c\| = 10$. After that, we sample three directions $v_1, v_2$ and $v_3$ with $\|v_i\| = 10$. Then we put three additional (outlier) clusters with means at $\mu_s + v_c + v_i$. This roughly corresponds to the right picture in Figure 4. The samples for each adversarial cluster are drawn from a distribution that matches the covariance of the inlier clusters, with the sample size being twice as large as of the affected inlier cluster.

2. *Adversarial line*: After sampling the inlier cluster means, we again choose the cluster with the smallest weight. Let $\mu_s$ denote its mean. Then, we sample a random direction $v_c$ with $\|v_c\| = 10$. We put three additional (outlier) clusters with means at $\mu_s + v_c, \mu_s + 2v_c$ and $\mu_s + 3v_c$, which form a line as shown in Figure 4. The samples are drawn similarly to the adversarial clusters, with the difference that the covariance is scaled by a factor of $5$ in the direction of the line.

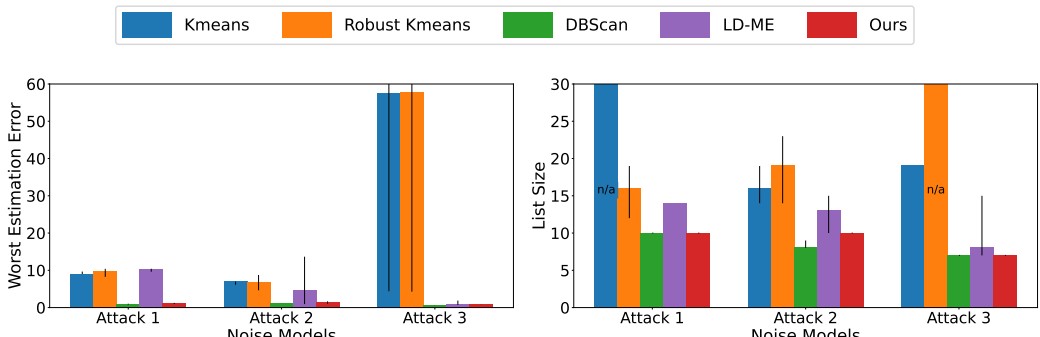

Figure 5: Comparison of five algorithms with three adversarial noise models. On the left we show worst estimation error of algorithms with constrained list size and on the right the smallest list size with constrained error guarantee. We plot the median of the metrics with the error bars showing 25th and 75th percentile. We observe that our method consistently outperforms prior works in terms of list size and worst estimation error, with the exception of DBSCAN, which performs at a similiar level.

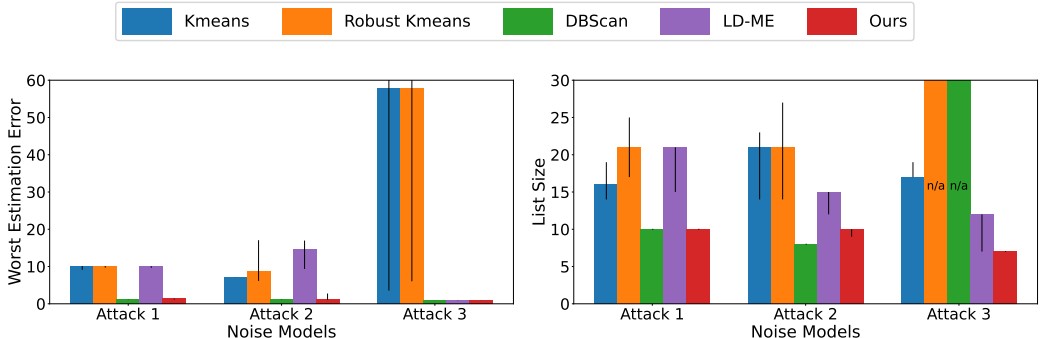

Figure 6: Worst estimation error and list size comparison for the case where inlier distributions are heavy-tailed. We can observe numerical stability of our approach.

3. *Gaussian adversary*: Here we simply introduce noise matching the empirical mean and covariance of all inlier data (i.e., as if all inlier clusters are generated from the same Gaussian distribution).

Note that in the first and second attack, the adversary creates clusters that do not respect the separation assumption of the true inlier clusters: either adversarial clusters are placed around the smallest inlier cluster (Adversarial Cluster), or the adversarial clusters form a line, pointing out in some fixed direction (Adversarial Line).

**Implementation details**  We implement the list-decodable mean estimation base learner in our InnerStage algorithm (Algorithm 3) based on [8]. It leverages an iterative multi-filtering strategy and one-dimensional projections. In particular, we use the simplified gaussian version of the algorithm. It is designed for distributions sampled from a Gaussian but also shows promising results for the experiments involving a heavy-tailed t-distribution as depicted in Figure 6. The robust mean estimator used to improve the mean hypotheses for large clusters is omitted in our implementation.

**Hyper-parameter search and experiment execution**  The hyper-parameters of our algorithm are tuned beforehand based on the experimental setup. For the comparative algorithms, hyper-parameter searches are conducted within each experiment after initial tuning. For our algorithm, key parameters include the pruning radius $\gamma$ used in the OuterStage routine (Algorithm 6) and $\beta$ used in the InnerStage (Algorithm 4). In addition, parameters for the LD-ME base learner, such as the cluster concentration threshold, also require careful selection, resulting in a total of 7 parameters. The tuning for these was performed using a grid search comparing about 250 different configurations. Similarly, we independently tune the vanilla LD-ME algorithm, which we run with $w_{\text{low}}$ as weight parameter. For

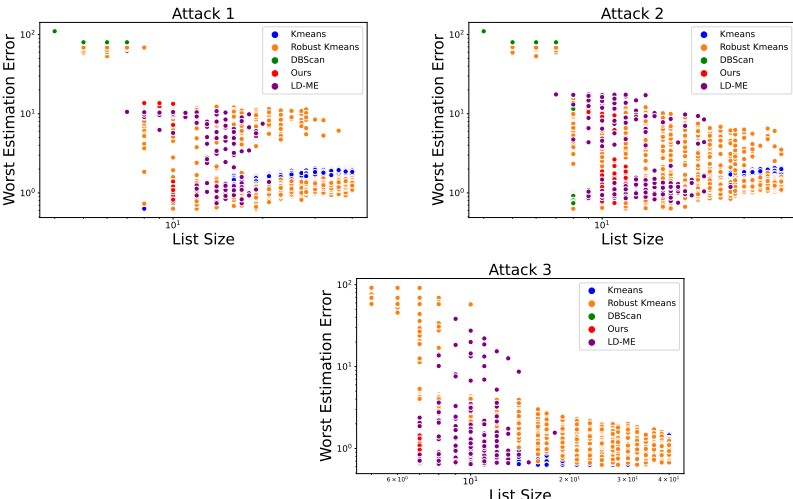

Figure 7: Scatter plot of all results for one iteration of the experiment using three adversarial noise models.

DBSCAN, we optimize the list size and error metrics by searching over a range of $100$ values for $\varepsilon$, which controls the maximum distance between samples considered in the same neighbourhood. The minimum samples threshold, which validates the density based clusters, is pretuned beforehand and adjusted based on $w_{\text{low}}$. For $k$-means and its robust version, utilizing a median-of-means weighting scheme, we explore $21$ values for $k$, including the true number of clusters. Each parameter setting is executed $100$ times to account for stochastic variations in the algorithmic procedures, such as $k$-means initialization. The list size and worst estimation error for each list of clusters obtained is visualized exemplarily for one iteration of the experiment in Figure 7. The plot provides insight into how the different algorithms perform and vary with different list sizes.

**Evaluation details**    Note that we have two sources of randomness: the data is random and also the algorithms themselves are random (except DBSCAN). For a clear comparison, we sample and fix one dataset for each attack model. we plot the performance of $100$ runs of each algorithm for each parameter setting, each time recording the returned list size together with the worst estimation error $\max_{i \in [k]} \min_{\hat{\mu} \in L} \|\mu_i - \hat{\mu}\|$. Then we either (i) report the worst estimation error for all runs with constrained list size (we pick the list size most frequently returned by our algorithm, specifically 7 or 10 in our experiments) (see Figure 5, left), or (ii) report the smallest list size required to achieve the same or smaller worst estimation error (we pick the 75th quantile of errors of our algorithm for a threshold) (see Figure 5, right). Under size constraint (i), the bar plots correspond to the median over the runs, with error bars indicating the 25th and 75th quantiles. Under error constraint (ii), the bar plots represent the minimum list size for which the median over the runs falls below the threshold, while the error bars show the minimum list size for which the 25th and 75th quantiles meet the constraint. Note that 'n/a' indicates that, within the scope of our parameter search, no list size achieves an error below the specified constraint.

In Figure 6 we study the numerical stability of our approach. In particular, whether the performance degrades when inlier distribution does not satisfy required assumptions. We observe that if one uses our meta-algorithm with base learner designed for Gaussian inliers, we still obtain stable results even in the case of heavy-tailed inlier distribution.

## I.1    Variation of $w_{\text{low}}$

To study the effect of varying $w_{\text{low}}$ input on the performance of our approach and LD-ME, we introduce a new noise model. As illustrated in Figure 8, we consider a mixture of $k = 3$ well-separated clusters: one small cluster with a weight of $0.045$ and two large clusters, each with a weight of $0.2$. We place two adversarial clusters (see paragraph on attack distributions for details): one near the small cluster and another near one of the large clusters. Furthermore, uniform noise is introduced,

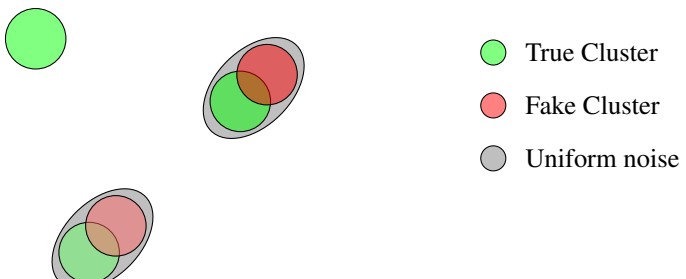

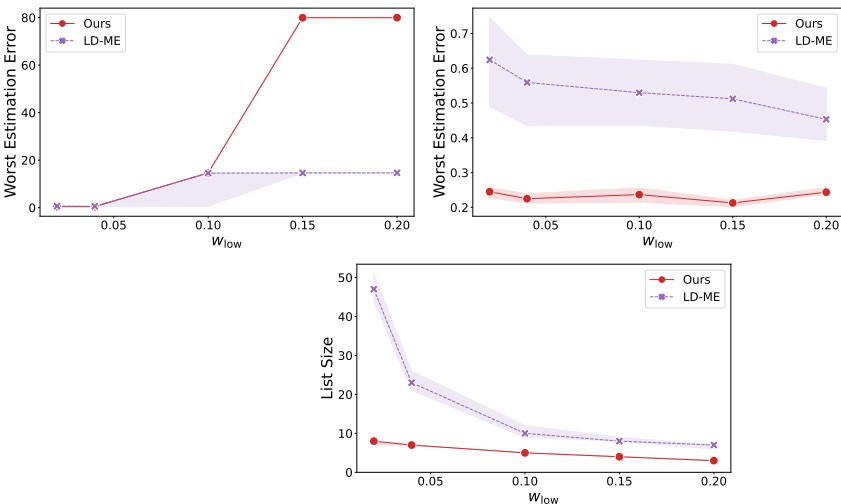

Figure 8: Setup for $w_{\text{low}}$ variation experiment with clusters contaminated by an adversarial cluster and uniform noise. Lower color intensities indicate smaller cluster weights.

Figure 9: Comparison of list size and estimation error for small and large inlier clusters for varying $w_{\text{low}}$ inputs. The experimental setup is illustrated in Figure 8. The plot on the top left shows the estimation error for the small cluster and the plot on the top right shows the error for the large cluster. We plot the median values with error bars indicating 25th and 75th quantiles. As $w_{\text{low}}$ decreases, our algorithm maintains a roughly constant estimation error for the large cluster, while the error for LD-ME increases.

spanning the range of the data generated by the inlier and its nearby outlier cluster and accounting for $10\%$ of the data in this region. Overall, $\varepsilon = 0.56$ and we draw 22650 samples from this mixture distribution.

For both algorithms we run 100 seeds for each $w_{\text{low}}$ ranging from 0.02 to 0.2, which corresponds to the weight of the largest inlier cluster. In Figure 9, we plot the median estimation error with error bars showing the 25th and 75th quantiles for the small cluster (top left) and the large cluster near the outlier cluster (top right). As expected from our theoretical results, we observe that our algorithm performs roughly constant in estimating the mean of the large cluster, regardless of the initial $w_{\text{low}}$. Meanwhile, the estimation error of LD-ME increases as $w_{\text{low}}$ decreases further below the true cluster weight. Furthermore, the plots show that our approach does consistently outperform LD-ME in terms of both worst estimation error and list size. Figure 10 also compares the performance of the clustering algorithms in this experimental setup with results similar to the ones obtained in the previous experimental settings.

## I.2    Computational resources

Our implementation of the algorithm and experiments leverages multi-threading. It utilizes CPU resources of an internal cluster with 128 cores, which results in a execution time of about 5 minutes for a single run of the experiment for one noise model with 10000 samples. We remark that classic

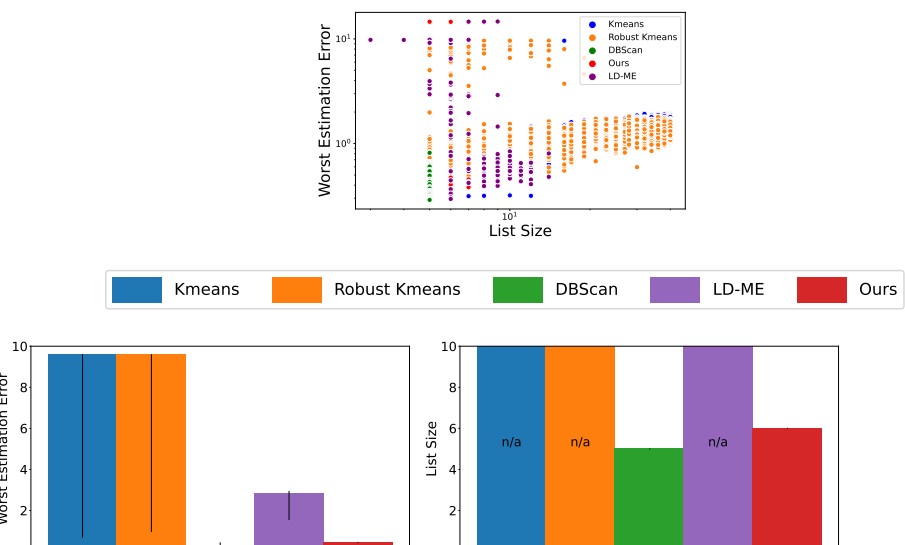

Figure 10: Worst estimation error and list size comparison for the setup used in the $w_{\mathrm{low}}$ variation experiment.

approaches like $k$-means and DBSCAN perform fast and the most time-consuming part is the execution of the LD-ME base learner. Given our experimental setup with three noise models, it takes about $15$ minutes to reproduce all our results for one data distribution.

