# OpenReview forum: "Robust Mixture Learning when Outliers Overwhelm Small Groups"
_NeurIPS.cc/2024/Conference — NeurIPS 2024 poster_

### Official Review · Reviewer_TTHY · 2024-07-12

**Soundness:** 3
**Presentation:** 3
**Contribution:** 3
**Rating:** 6
**Confidence:** 3

**Summary:**

In this paper, the authors introduce the list-decodable mixture learning problem, which can be considered as the extension of the list-decodable mean estimation problem. In such case, data is drawn from a weighted mixture of $k$ inlier distributions and an adversarial outlier distribution. A notable aspect of this paper’s problem setting is that the fraction of outlier $\varepsilon$ can exceed the  fraction of certain inlier clusters $w_i$s. As a result, the required list size is at least $k+\varepsilon/\min w_i$. In this paper, The primary objective of this study is to compute a compact list $L$ of means such that for any $\mu_i$ of an inlier group, there exists $\hat{\mu}\in L$ with a sufficiently small $\| \hat{\mu}-\mu_i \|$. The only known knowledge is $w_{low}$, a lower bound of all $w_i$s.

The algorithm proposed by the author is meta in the sense that it uses other algorithms (Robust Mean Estimation and List-Decodable Mean Estimation) as base components. It comprises two stages outlined as follows:

1. outer stage: Iteratively separates the inlier clusters from each other and from the outliers. This stage produces the collection of sets $\mathcal{T}$ satisfying certain properties. Then for every $T\in \mathcal{T}$, run the inner stage.
2. Inner stage: uses the cor-kLD algorithm to derive the cor-aLD algorithm that restricts access to the $\alpha_{low}$ s.t. $\alpha_{low}\leq \alpha$. In this stage, for every $T\in \mathcal{T}$, a refined list of mean estimates is produced.

**Strengths:**

- This study is interesting. It appears to be the first one to address the challenge of mixture learning in cases where outliers overwhelm small inlier groups
- The problem setting of this paper is practical. The algorithm is designed to work with limited information, only having access to the lower bound of inlier group fractions.
- The paper presents theoretical lower bounds for the error. Remarkably, the proposed meta-algorithm achieves a matching lower bound.

**Weaknesses:**

- The two stage meta-algorithm can be time-consuming.

**Questions:**

- The paper lacks a presentation of the time complexity, and the experimental part also fails to provide results from this aspect. It would be beneficial to include information regarding the time complexity of the proposed algorithm to provide a comprehensive understanding of its computational efficiency.

---

> ### Author Rebuttal · Authors · 2024-08-07
>
> We thank the reviewer for their appreciation of our work and for the feedback which will help to improve it. Below, we address specific comments and questions.
>
> > The two stage meta-algorithm can be time-consuming. [...] The paper lacks a presentation of the time complexity, and the experimental part also fails to provide results from this aspect. It would be beneficial to include information regarding the time complexity of the proposed algorithm to provide a comprehensive understanding of its computational efficiency.
>
> Thank you for your comment. We provided a detailed analysis of the time complexity in the general rebuttal response (third part). In particular, our meta-algorithm has a small overhead complexity compared to base learners. We will include this discussion in the revised version.

---

> > ### Comment · Reviewer_TTHY · 2024-08-12
> >
> > Thank you for the additional analysis. The authors have answered my question and I would like to keep my postive score.

---

### Official Review · Reviewer_hCCi · 2024-07-20

**Soundness:** 3
**Presentation:** 3
**Contribution:** 3
**Rating:** 6
**Confidence:** 3

**Summary:**

This paper addresses the problem of estimating the means of well-separated mixtures in the presence of adversarial outliers, a scenario where traditional algorithms may fail. The authors introduce the concept of list-decodable mixture learning (LD-ML), which is particularly relevant when outliers can outnumber smaller inlier groups.

**Strengths:**

The paper introduces a new problem formulation (LD-ML) and a creative combination of existing ideas (base learners for adversarial corruptions) to solve it. This originality addresses significant limitations in prior work, enhancing the robustness and applicability of mixture learning algorithms. The research quality is high, with rigorous theoretical analysis and well-designed experiments. The thoroughness of the proofs and the clarity of the theoretical contributions reflect a strong understanding of the subject matter.

**Weaknesses:**

The paper did not include real-world datasets and additional robust learning methods for comparison. It would be great if the impact of the assumptions can be clarified.

**Questions:**

1. Have you considered validating your algorithm on real-world datasets? If so, could you provide details on these experiments? If not, what were the main obstacles?It would be beneficial to include results from real-world datasets to demonstrate the practical utility and robustness of your method.

2. How does your algorithm perform when the assumption of well-separated mixture components does not hold? Have you tested its robustness in such scenarios?

3. Can you provide a detailed analysis of the computational complexity of your algorithm? How does it scale with the number of dimensions and mixture components?

4. How does your algorithm compare to other state-of-the-art robust learning methods in terms of practical performance metrics like runtime and ease of implementation?

---

> ### Author Rebuttal · Authors · 2024-08-07
>
> We thank the reviewer for their appreciation of our work and for the feedback which will help to improve it. Below, we address specific comments and questions.
>
> **Experimental evaluation**
>
> > The paper did not include real-world datasets and additional robust learning methods for comparison. [...] Have you considered validating your algorithm on real-world datasets? If so, could you provide details on these experiments? If not, what were the main obstacles? It would be beneficial to include results from real-world datasets to demonstrate the practical utility and robustness of your method. [...] How does your algorithm perform when the assumption of well-separated mixture components does not hold? Have you tested its robustness in such scenarios?
>
> We would like to emphasize that the focus of our work is to first introduce the LD-ML framework and prove theoretical guarantees in this setting for an efficient algorithm. We agree that for the purpose of properly evaluating the effectiveness of the algorithm in practice we would need a much more extensive experimental setup that would be out of scope for this paper and we leave it for future work.
>
> Having said that, it would be great if the reviewer could point to the specific robust learning methods (beyond the ones considered in the paper) they have in mind. For our experimental results, we opted for comparing our algorithm with 1) baselines that have provable guarantees in our setting (list decoding algorithm) and 2) three commonly used in practice clustering algorithms: (k-Means, robust k-Means and DBSCAN).
>
> >  It would be great if the impact of the assumptions can be clarified. [...] How does your algorithm perform when the assumption of well-separated mixture components does not hold? Have you tested its robustness in such scenarios?
>
>
> We will clarify the assumptions (in particular, we have a result which does not require well-separateness assumption) in the revised version. For example, Corollary C.4 answers the question about the impact of the well-separateness assumption, by showing the guarantees when the components are non-separated. We can move it to the main text so that the impact is more explicit in the main text.
>
> Furthermore, in our experimental design we effectively included the setting where clusters are not separated (by inserting 'adversarial' clusters in the vicinity of the inlier component). Therefore, results in Figure 2 can also be interpreted as results for non-separated clusters (by assuming that the inserted clusters around the smallest cluster are part of the mixture).
>
> > Can you provide a detailed analysis of the computational complexity of your algorithm? How does it scale with the number of dimensions and mixture components?
>
> Thank you for your suggestion. Please see our general rebuttal response (third part) for the detailed analysis of the computational complexity.

---

> > ### Comment · Reviewer_hCCi · 2024-08-12
> >
> > Thank you for the response. I am happy to raise my score considering all the discussions.

---

### Official Review · Reviewer_w5FP · 2024-07-22

**Soundness:** 4
**Presentation:** 4
**Contribution:** 4
**Rating:** 8
**Confidence:** 2

**Summary:**

The authors investigate the problem of mean estimation for a well-separated mixture in the presence of arbitrary outliers introduced by an adversary. They propose a meta-algorithm that leverages robust mean estimation algorithms as base learners, each with a set of prescribed properties. The authors provide an error guarantee of $\mathcal{O}(\sqrt{\log \frac{\epsilon}{w_i}})$ for a list size of $k + \mathcal{O}(\frac{\epsilon}{w_{low}})$, where $k$ is the total number of inlier groups, $\epsilon$ is the proportion of outliers, and $w_i$ is the proportion of inlier group $i$. Additionally, their approach is capable of handling cases where some $w_i \leq \epsilon$.

**Strengths:**

The main contribution of the paper lies in developing the meta-algorithm (Algorithm B.1) which first creates clever partitions of the original set (Algorithm D.1) and then uses the appropriate base learners on the partitioned sets (Algorithm C.1). While I have not gone through the mathematical details very carefully, the results seem very interesting and match the error guarantees of base learners run with the full information on the weight proportion of each inlier group separately. The paper is written well and the ideas and corresponding arguments are presented clearly in the main paper with technical details deferred to the appendix.

**Weaknesses:**

The paper makes significant contributions, and I did not identify any particular weaknesses. While I possess general knowledge of the research area, I am not an expert, and my review should be considered within that context.

**Questions:**

Please see above.

**Limitations:**

The limitations are adequately addressed.

---

> ### Author Rebuttal · Authors · 2024-08-07
>
> We thank the reviewer for their appreciation of our work.

---

### Official Review · Reviewer_fDjD · 2024-07-27

**Soundness:** 3
**Presentation:** 2
**Contribution:** 3
**Rating:** 6
**Confidence:** 3

**Summary:**

This submission considers the problem of list-decoding the means of a mixture of $k$ sub-Gaussian components, under adversarial *additive* contamination which can have size $\epsilon n$ larger than the smallest-weight component. Instead of only yielding guarantees scaling with the known component weight lower bound $\alpha_{low}$, the (meta-)algorithm in this paper achieves error that depends directly on $w_i$, the weight of that particular component, and decays also with $\epsilon$. Furthermore, the size of the output list is guaranteed $k + O(\epsilon/\alpha_{low})$ instead of the more common and weaker $O(1/\alpha_{low})$ guarantee.

Note: this is an emergency review, so I did not read the paper as closely and carefully as I'd like.

**Strengths:**

To me, the main strength of the paper lies in identifying the regimes where much stronger guarantees (e.g. mean estimation error dependence on actual component weight and not just $\alpha$) can be made compared to prior works, which are worst-case optimal in some sense of the phrase. The (meta-)algorithmic ideas are also simple enough to be implemented (which I consider a plus and not a minus).

**Weaknesses:**

- The introduction claims results even when there is no separation between mixture components, but Theorem 3.3 makes a separation assumption? Did I miss something?

====

Below are hopefully some constructive feedback that the authors can use to improve the paper:

- These guarantees seem possible only because of the distance separation vs concentration assumption (both depend on "$t$"), and I think there needs to be more discussion of the intuition on the quantitative tradeoff in the main body.

- The contexualization in prior work seems incomplete to me. I have two related complaints, both (roughly) related to finite covariance assumption setting.

1. The $\epsilon \gg \min_i w_i$ regime seems to me *almost* covered by the standard setting of learning the means of finite covariance mixtures: the contamination distribution $Q$ can always be trimmed a little bit to get finite covariance, and just be regarded as part of the mixture. Upon googling, it seems that there is more recent work by Diakonikolas, Kane, Lee and Pittas (Clustering Mixtures of Bounded Covariance Distributions Under Optimal Separation) which does handle such mixtures, and in fact components with different covariance sizes. I guess the caveat is that, under the above modelling, the mean of $Q$ might not be sufficiently bounded away from the genuine components. Is my understanding correct? I think a comparison to finite-covariance mixture learning literature, from the framework perspective, would be helpful.

2. The intro of this submission really focuses on the sub-Gaussian mixture case ($t \approx \log 1/w_{low}$), but the general result applies also to $t = 2$. So I think a direct comparison to the technical results of the above paper as well as DKKLT22 is needed. For example, the separation assumption in this submission is much larger ($1/w_{low}^2$) than these prior works ($1/\sqrt{w_{low}}$). In general, I think it'd be very useful to clarify the results in this paper with respect to these DKKLT22 and DKLP papers (and other related works).

- Writing: Section 3.1 was quite dense to read (though possibly because I have to read it quickly for the emergency review). It might be better to present the Gaussian case and the relation to prior work, before presenting the full result in generality and explaining precisely the generalizations.

- Minor thing in line 114: I'm not sure it's strictly true that "in most robust estimation problems, the fractions of inliers and outliers are usually provided to the algorithm". Doesn't filtering work even if the number of outliers is unknown, as long as the inlier variance is known? One could keep filtering until the remaining data set has small-enough covariance?

**Questions:**

- Any intuition for why $w_{low}^2$ is the right quantity to compare with $w_i$ and $\epsilon$? Is it a necessity of the problem setting or is it just an artifact of the algorithmic construction?

- Definition 2.1 is essentially assuming boundedness compared to the identity covariance Gaussian. For mixtures where the components have, say, covariance bounded by $c$ times identity, does the algorithm need to know $c$, or can $c$ be estimated easily in this context?

- How hard is it to go beyond the corruption model of (2.1), to get *adaptive* additive corruption? That is, the corruption isn't just drawn from a fixed $Q$, but that the corrupted points can depend directly on the drawn inlier samples?

**Limitations:**

Yes

---

> ### Author Rebuttal · Authors · 2024-08-07
>
> We thank the reviewer for their appreciation of our work and for the feedback which will help to improve it. Below, we address specific comments and questions.
>
> > The introduction claims results even when there is no separation between mixture components, but Theorem 3.3 makes a separation assumption? Did I miss something?
>
> Thank you for the comment. Yes, we also obtain a result for the non-separated case. See lines 241-243 in the main text, and Corollary C.4 in the Appendix. In the revised version we will move the corollary statement to the main text.
>
> > These guarantees seem possible only because of the distance separation vs concentration assumption (both depend on $t$), and I think there needs to be more discussion of the intuition on the quantitative tradeoff in the main body.
>
> Indeed, this trade-off is inevitable and is present in prior works too. Please also see our discussion on the separation assumption in the general rebuttal (second part).
>
> > Upon googling, it seems that there is more recent work by Diakonikolas, Kane, Lee and Pittas (Clustering Mixtures of Bounded Covariance Distributions Under Optimal Separation) which does handle such mixtures, and in fact components with different covariance sizes. I guess the caveat is that, under the above modelling, the mean of $Q$ might not be sufficiently bounded away from the genuine components. Is my understanding correct? I think a comparison to finite-covariance mixture learning literature, from the framework perspective, would be helpful.
>
> Your understanding is correct, after the trimming operation, $Q$ is not guaranteed to be separated from the other components. This does not allow to model $Q$ as a part of the mixture.
> We will add a comparison with the DKLP work in the revised version (also see general rebuttal response).
>
> > The intro of this submission really focuses on the sub-Gaussian mixture case ($t \approx \log⁡ 1 / w_{\text{low}}$), but the general result applies also to $t = 2$. So I think a direct comparison to the technical results of the above paper as well as DKKLT22 is needed. For example, the separation assumption in this submission is much larger $1 / w_{\text{low}}^2$ than these prior works $1 / \sqrt{w_{\text{low}}}$. In general, I think it'd be very useful to clarify the results in this paper with respect to these DKKLT22 and DKLP papers (and other related works).
>
> Thank you for the comment, please see our discussion on the separation assumption in the general rebuttal (second part), where we also compare with [DKLP23]. We will extend the comparison with prior work in the revised version.
>
> > It might be better to present the Gaussian case and the relation to prior work, before presenting the full result in generality and explaining precisely the generalizations.
>
> Thank you for your suggestion, we will take this into consideration for the revised version.
>
> > I'm not sure it's strictly true that "in most robust estimation problems, the fractions of inliers and outliers are usually provided to the algorithm". Doesn't filtering work even if the number of outliers is unknown, as long as the inlier variance is known? One could keep filtering until the remaining data set has small-enough covariance?
>
> Thank you for your comment, this was indeed an imprecise choice of words.
> We were mostly referring to the list decoding setting, where algorithms are provided with inlier proportion (see, e.g., [1, 2]). Here, the extension to the unknown inlier proportion seems challenging for the following two reasons:
> First, e.g., for the filtering technique, in order to obtain optimal error, one needs to filter with the ‘correct’ polynomial degree, and this degree depends on the fraction of inliers.
> Second, in order to obtain a list of predictions, a lower bound on the size of inlier sets is required, in order to decide when to ‘discard’ too small subsets.
> Overall, we argue that it is not straightforward to extend prior methods to unknown component weights, and this extension is one of our contributions.
>
> > Any intuition for why $w_{\text{low}}^2$ is the right quantity to compare with $w_i$ and $\varepsilon$?
>
> Thank you for the question. First, we would like to note that it is generally not possible to obtain guarantees depending only on $w_i / (w_i + \varepsilon)$, since there are always samples from other components which are effectively ‘outliers’ for a fixed component. For example, consider the Gaussian case with $\varepsilon = 0$, and $w_i / (w_i + \varepsilon) = 1$, but the approximation error is clearly greater than $0 = \sqrt{\log (w_i + \varepsilon) / w_i)}$.
>
> Further, the specific value $w_{\text{low}}$ connects to the separation assumption (see discussion on the necessity of the separation assumption in the general response): if the components are $\Omega\left(\left(1 / w_{\text{low}}\right)^{4/t}\right)$ separated, only an $O(w_{\text{low}}^4)$ fraction of samples from one mixture component may lie in the vicinity of another, so in total $O(w_{\text{low}}^3) \leq w_{\text{low}}^2$ points from other components. We also remark that $w_i$ always dominates $w_{\text{low}}^2$.
>
>
> > Definition 2.1 is essentially assuming boundedness compared to the identity covariance Gaussian. For mixtures where the components have, say, covariance bounded by 𝑐 times identity, does the algorithm need to know 𝑐, or can 𝑐 be estimated easily in this context?
>
> We assume that the algorithm knows at least a valid upper bound (and obtain guarantees depending on this upper bound). We leave the question of unknown covariances for future work (the main challenge here is to still obtain a small list size $k + O(\varepsilon / w_{\text{low}})$, while correctly ‘guessing’ the covariance).
>
> > How hard is it to go beyond the corruption model of (2.1), to get adaptive additive corruption?
>
> Thank you for the question, please see our general rebuttal response (first part).
>
> ---------
> For references [1, 2], please see the main rebuttal response.

---

> > ### Comment · Reviewer_fDjD · 2024-08-08
> >
> > Thanks for your response. More comments from me:
> >
> > - Re: $1/w_{low}^{4/t}$. For the $1/\sqrt{w_{low}}$ kind of separation, for bounded covariance mixtures, there is a simple intuition of Chebyshev's inequality to see that it is indeed necessary (and hence tight up to constants). Is there any analogously intuitive explanation for $1/w_{low}^{4/t}$? That's what I was hoping for. The overall response doesn't quite get at that level of intuition.
> >
> > - On the comparison of $w_{low}^2$ with $w_i$ and $\epsilon$, it seems from the explanation that it is slightly arbitrary (it could've been anything bigger than $w_{low}^3$?). In that case, it might be worth pointing out in the paper, for intuition for the reader.
> >
> > I'll reply to the overall rebuttal thread for my comments on the comparison with bounded-covariance mixture clustering works.

---

> > > ### Author Response · Authors · 2024-08-11
> > >
> > > > Re: $1 / w_{\text{low}}^{4/t}$. For the $1 / \sqrt{w_{\text{low}}}$ kind of separation, for bounded covariance mixtures, there is a simple intuition of Chebyshev's inequality to see that it is indeed necessary (and hence tight up to constants). Is there any analogously intuitive explanation for $1 / w_{\text{low}}^{4/t}$? That's what I was hoping for. The overall response doesn't quite get at that level of intuition.
> > >
> > > Thank you for the comment. It is true that when $t=2$ and $\varepsilon \lesssim w_{\text{low}}$ the optimal separation is $O(1 / w_{\text{low}}^{1/t})$ (we note that this was achieved with optimal list size only last year). However, as far as we can tell, this separation is currently known to be achievable only when $t = 2$ and $\varepsilon \lesssim w_{\text{low}}$. In particular,
> > > 1. When $t \geq 4$, to the best of our knowledge, the best mixture learning algorithm needs separation $1 / w_{\text{low}}^{2/t}$, even when there are no outliers [e.g., Kothari-Steinhardt-Steurer’18, Theorem 2.7]. This is still better than ours, but larger than $1 / w_{\text{low}}^{1/t}$.
> > >
> > > 2. When $\varepsilon \gg w_{\text{low}}$, as far as we can tell, nothing is known about the separation required to obtain optimal error and list-size. Our paper is the first in this setting.
> > >
> > > Our goal was to obtain an algorithm that works when $\varepsilon \gg w_{\text{low}}$, and we focused on $t$-sub-Gaussian moments to show the generality of our result (importantly, with an application to clustering Gaussians). In this setting it is unclear that separation $O(1 / w_{\text{low}}^{1/t})$ is possible, given the observations in (1) and (2). It is possible that a much tighter analysis of our techniques may allow separation $O(1 / w_{\text{low}}^{2/t})$; we did not consider this kind of optimization essential to our paper, but acknowledge that it would be important future work to gain an even deeper understanding of the problem.
> > >
> > > On a high level, the separation is currently used in our proof for the following technical reason:
> > > a separation of $1 / w_{\text{low}}^{1/t}$ suffices to show that the samples from a component are concentrated well when projected on a given direction. Our analysis needs such a property to hold along $1/w_{\text{low}}^2$ directions (between all pairs of components). In particular, in Lemma G.2, when applied for number of directions $m = 1 / w_{\text{low}}^2$, we need the separation $R$ to be at least $1 / w_{\text{low}}^{2/t}$ for a union bound to be applicable. Furthermore, to prove a technical Lemma D.6 we end up needing separation $1 / w_{\text{low}}^{4/t}$.
> > >
> > > > On the comparison of $w_{\text{low}}^2$ with $w_i$ and $\varepsilon$, it seems from the explanation that it is slightly arbitrary (it could've been anything bigger than $w_{\text{low}}^3$?). In that case, it might be worth pointing out in the paper, for intuition for the reader.
> > >
> > > Indeed, the reviewer is correct that, assuming separation $1 / w_{\text{low}}^{4/t}$, anything larger than $C w_{\text{low}}^3$, for some constant $C > 0$ could be used. We will add this clarification to the revised version.
> > >
> > >
> > > [Kothari-Steinhardt-Steurer’18] Kothari, Pravesh K., Jacob Steinhardt, and David Steurer. "Robust moment estimation and improved clustering via sum of squares." Proceedings of the 50th Annual ACM SIGACT Symposium on Theory of Computing. 2018.

---

> > > > ### Comment · Reviewer_fDjD · 2024-08-11
> > > >
> > > > Thanks for the responses. I'm happy with this discussion and hope the authors will incorporate this into the final paper. I've raised my score accordingly.

---

### Official Review · Reviewer_izza · 2024-07-27

**Soundness:** 3
**Presentation:** 3
**Contribution:** 3
**Rating:** 7
**Confidence:** 3

**Summary:**

The paper studies the problem of estimating the means of a mixture model in the presence of outliers. More specifically, each component has unknown weight $w_i$ that is lower bounded by a known quantity $w_{\mathrm{low}}$, each component is assumed to be (sub)-Gaussian, and the mixture also includes an adversarially selected distribution (outliers) with weight $\epsilon$, which can in general be larger than $w_{\mathrm{low}}$, meaning that the outliers can form entire clusters on their own. Correctly estimating the means of all true components is impossible and thus the goal here is to instead output a small list of candidate means. Since the $w_i$’s are unknown, naive applications of existing list-decoding algorithms would result in estimation errors and list size guarantees that are a function of $w_{\mathrm{low}}$ only. The contribution of this paper is to show that it is indeed possible to obtain more fine-grained guarantees where the error for the $i$-th component scales with $w_i$ instead of $w_{\mathrm{low}}$ and the list size is equal to the true number of components plus a small overhead $O(\epsilon/w_{\mathrm{low}})$. Regarding the list-size, $\epsilon/w_{\mathrm{low}}$ is the number of extra components that the outliers can form, thus this term is unavoidable in the list size. The paper provides information-theoretic error lower-bounds justifying that the precise error rates achieved are qualitatively tight. The regimes where the final algorithm has qualitatively best-possible performance includes the cases where the means are pairwise separated as well as without that assumption, and also the case where $\epsilon \ll w_0$ for separated mixtures (where it matches existing qualitatively optimal algorithms).

The algorithm (for the separated components case) in based on the following parts shown in the paper: First, there is a way to obtain list-decodable mean estimation algorithms that do not need knowledge of the fraction of inliers. This works by collecting the answers of a list-decodable mean estimator for multiple candidate values for the fraction of inliers and carefully pruning the results to keep only hypothesis with a large number of points close to them. Second, the paper proposes a procedure that splits the dataset into (not too many) parts where each part includes at most one (almost entire) cluster of inliers, so that calling the agnostic list-decodable mean estimation of the previous sentence can produce the improved error guarantees. Finally, for the case where $\epsilon \ll w_{\mathrm{low}}$, existing robust mean estimation algorithms can be employed to further improve the error.

Overall, the paper closes a gap in the literature. The result is non-trivial and a useful addition to the literature, thus I am recommending acceptance.

**Strengths:**

* A positive point of the algorithm is that it is a meta-algorithm, in the sense that only performs calls to existing algorithms (for list-decoding and robust mean estimation), and performs computationally simple processing of the dataset based on the outputs of these algorithms. The algorithm is thus efficient and takes advantage of computational efficiency of the base learners.

* The paper contains some experiments to demonstrate practical performance advantages.

**Weaknesses:**

* The contamination model of eq (2.1) requires that corruptions come i.i.d. from some distribution. Since the base learners can also handle some adversarial corruptions why can’t the final algorithm work for fully adversarial contamination?

* On presentation: The result about non-separated mixtures is not discussed in the main body. It would be good to at least provide the main ideas regarding why similar error can be achieved for the non-separated case, since the algorithm discussed in the main body seems to use crucially the separation of components. Another slightly confusing point is that the introduction talks about Gaussianity a lot while Theorem 3.3 works even for bounded second moment distributions. Existing black box learners work for these settings too but for some reason the existence of black box learners is emphasized only for the Gaussian case (line 193) in Section 3.1.

**Questions:**

* The algorithm that gets errors all the way down to $O(\sqrt{\log(1/w_i})$ has super-polynomial complexity. Does the existing SQ lower bound from list-decodable mean estimation in [3] justify this via some easy reduction? If so, it would be good to include.

* Is the optimal separation for the bounded moments case indeed $O(1/w_{\mathrm{low}}^{4/t})$? I just want to make sure that there is no typographical error in the exponent, since for bounded second moments usually things scale with square root of $w_\mathrm{low}$.

* Also see first point in "weaknesses".

**Limitations:**

-

---

> ### Author Rebuttal · Authors · 2024-08-07
>
> We thank the reviewer for their appreciation of our work and for the feedback which will help to improve it. Below, we address specific comments and questions.
>
> > The contamination model of eq (2.1) requires that corruptions come i.i.d. from some distribution. Since the base learners can also handle some adversarial corruptions why can’t the final algorithm work for fully adversarial contamination?
>
>
> Thank you for the question, please see our general rebuttal response (first part).
>
> > On presentation: The result about non-separated mixtures is not discussed in the main body. It would be good to at least provide the main ideas regarding why similar error can be achieved for the non-separated case, since the algorithm discussed in the main body seems to use crucially the separation of components.
>
> Thank you for the suggestion, we will mention the result on non-separated mixtures in the main text in the revised version.
>
> > Another slightly confusing point is that the introduction talks about Gaussianity a lot while Theorem 3.3 works even for bounded second moment distributions. Existing black box learners work for these settings too but for some reason the existence of black box learners is emphasized only for the Gaussian case (line 193) in Section 3.1.
>
> Thank you for the suggestion, we will extend the discussion on the results for the bounded moment distributions in the revised version.
>
> > The algorithm that gets errors all the way down to $𝑂(\sqrt{\log⁡(1/w_i)})$ has super-polynomial complexity. Does the existing SQ lower bound from list-decodable mean estimation in [3] justify this via some easy reduction? If so, it would be good to include.
>
> Yes, there is a simple reduction of SQ lower bounds (same as IT lower bounds), we mentioned it in the Appendix (lines 1001-1004). We will move the paragraph to the main text in the revised version.
>
> > Is the optimal separation for the bounded moments case indeed $O\left(\left(\frac{1}{w_{\text{low}}}\right)^{4/t}\right)$? I just want to make sure that there is no typographical error in the exponent, since for bounded second moments usually things scale with square root of $w_{\text{low}}$.
>
> Thank you for raising this point. Yes, there is no typographical error (please see our general rebuttal response, second part).

---

> > ### Comment · Reviewer_izza · 2024-08-12
> >
> > Thank you for the clarifications. I am keeping my positive score.

---

### Author Rebuttal · Authors · 2024-08-07

We thank all reviewers for their valuable feedback and comments, which help to improve our manuscript. Below, we focus on three main concerns, which were raised by several reviewers.

**Adaptive contamination**.
We prove robustness of our algorithm in the non-adaptive contamination model, where $\varepsilon$ proportion of data are i.i.d. samples from an adversarial distribution $Q$. The reviewers asked whether our results can be applied to adaptive (non-i.i.d.) contaminations, or whether this is a limitation of our method.
The proof of Theorem 3.3 only uses the concentration of inlier samples, and thus generalizes to the case of adaptive contaminations. However, recall that the guarantees of our meta-algorithm depend on the base learner guarantees. Therefore, as long as both base learners (RME and LD-ME) have guarantees under adaptive contaminations, so does our meta-algorithm.

For RME, state-of-the-art methods are indeed robust against adaptive adversaries. However, to the best of our knowledge, results for LD-ME (see, e.g., [1, 2]) are generally only stated for the non-adaptive contamination model. This limits our result to only i.i.d. contaminations.


**Optimality of separation requirements.**
In our paper, we present two sets of results: for non-separated (in the Appendix, Corollary C.4) and separated mixtures (Theorem 3.3, Corollary 3.4). In the former case, we note that for the smallest component, our algorithm has optimal guarantees. In particular, it is impossible to achieve asymptotically smaller error (even when $\varepsilon = 0$) unless the algorithm outputs an exponentially larger list size (see lines 238 - 243).
For separated components, we discuss the Gaussian distribution and distributions with sub-Gaussian $t$-th central moments separately.
For the case of Gaussian components, our separation requirements are $\Omega\left(\sqrt{\log 1 / w_{\text{low}}}\right)$, matching the information-theoretical lower bound for the separation (see [1]).

For general $t$, we require separation $\Omega\left(\left(1 / w_{\text{low}}\right)^{4 / t}\right)$ and the reviewers are correct in pointing out that there exist recent prior works on clustering of mixture models which require only separation $\Omega\left(\left(1 / w_{\text{low}}\right)^{1 / t}\right)$ (see, e.g., [3]). However, e.g., in [3], this comes at the expense of a larger list size $O(1 / w_{\text{low}})$ (see Theorem 3.1 in [3]).
Next, we provide intuition why, for a desired short list size $k + O(\varepsilon / w_{\text{low}})$, obtaining results under separation $\Omega\left(\left(1 / w_{\text{low}}\right)^{1 / t}\right)$ is challenging:
Note that at this separation, only a constant fraction of initial mass of the cluster stays ‘close’ to the cluster mean; the other constant fraction will be further away from the mean than $O\left(\left(1 / w_{\text{low}}\right)^{1 / t}\right)$. For our goal to prove the list size bound $k + O\left(\varepsilon / w_{\text{low}}\right)$, we require that such ‘left-over’ samples are not misinterpreted by the algorithm as separate clusters (see lines 674 - 697, the proof of Lemma B.7). Otherwise, we cannot guarantee list size better than $O\left(1 / w_{\text{low}}\right)$, which can be much larger than $k + O\left(\varepsilon / w_{\text{low}}\right)$.

Overall, the reason for our separation requirements is that
1. We consider the setting with a large adversarial part, which was not present in the literature before,
2. We guarantee optimal list size $k + O\left(\varepsilon / w_{\text{low}}\right)$.

We will add a detailed discussion on the separation assumption in the revised version and leave the very interesting question of relaxing the separation assumption for future work.

**Time complexity.**
Several reviewers asked for a detailed exposition of the computational complexity of our meta-algorithm. We would like to highlight that the main purpose of our work is to show the existence of a (quasi-)polynomial-time meta-algorithm with the proven performance guarantees. Our meta-algorithm uses RME and LD-ME base learners, thus it depends on their time complexity, and we did not optimize for the particularly fast and iteration-efficient implementation of our algorithm.
For the rebuttal we provide the following upper bounds on the time complexity (ignoring the RME base learner, which is generally faster than LD-ME learner):
$$\text{Inner stage:} \quad \tilde O\left(\left(\frac{1}{w_{\text{low}}}\right) T(n, w_{\text{low}}) + \left(\frac{1}{w_{\text{low}}}\right)^3 n\right),$$ where $T(n, w_{\text{low}})$ is the time for LD-ME base learner to run on a dataset with $n$ samples and $w_{\text{low}}$ fraction of inliers.
$$\text{Outer stage:} \quad O\left( T(n, w_{\text{low}}) + \left(\frac{1}{w_{\text{low}}}\right)^2 n\right).$$
$$\text{Full algorithm:} \quad \tilde O\left(\left(\frac{1}{w_{\text{low}}}\right) T(n, w_{\text{low}}) + \left(\frac{1}{w_{\text{low}}}\right)^4 n\right).$$
We leave the question of more efficient implementations for future work.

[1] Diakonikolas, Ilias, et al. "List-decodable robust mean estimation and learning mixtures of spherical gaussians." Proceedings of the 50th Annual ACM SIGACT Symposium on Theory of Computing. 2018.
[2] Cherapanamjeri, Yeshwanth, et al. "List decodable mean estimation in nearly linear time." 2020 IEEE 61st Annual Symposium on Foundations of Computer Science (FOCS). IEEE, 2020.
[3] Diakonikolas, Ilias, et al. "Clustering Mixtures of Bounded Covariance Distributions Under Optimal Separation." arXiv preprint arXiv:2312.11769 (2023).

---

> ### Comment · Reviewer_fDjD · 2024-08-08
>
> I skimmed the (intro of the) DKKLT22 and DKLP23 papers again. The latter seems to have some special-case results where they return a list of exactly $k$ components too, under extra assumptions like weight-uniformity or something about "no large sub-clusters". So it seems that this present submission vs these other works are making different assumptions and getting different results, and this work doesn't subsume the other works as the rebuttal seems to suggest. This is absolutely fine, of course, but it should be clearly explained in the revised submission given the connections between the different lines of work.

---

> > ### Author Response · Authors · 2024-08-11
> >
> > Thank you for your comment. Indeed, our main case of study is the possibly non-uniform mixture with $\varepsilon \gtrsim w_{\text{low}}$, where we obtain novel results. We will clarify the connections and differences from the prior works in the revised version.

---

### Decision · Program_Chairs · 2024-09-25

**Decision:**

Accept (poster)

**Comment:**

This paper provides novel algorithmic guarantees for an appropriate notion of clustering mixtures of separated structured distributions, including Gaussians. The work can handle the outlier-robust case where the fraction of outliers can be larger than the smallest cluster, at the expense of outputting a list of candidates for the means of the clusters. Overall, the reviewers agreed that this is an interesting contribution. Detailed discussion with a couple of reviewers revealed that additional comparison (and missing citations) to prior work would make the contributions of this work more transparent. The authors are encouraged to address these points in the revised version of their paper.